# Vertex sliding drives intercalation by radial coupling of adhesion and actomyosin networks during *Drosophila* germband extension

Timothy E Vanderleest[1†], Celia M Smits[2†], Yi Xie[2], Cayla E Jewett[2], J Todd Blankenship[2]*, Dinah Loerke[1]*

[1]Department of Physics and Astronomy, University of Denver, Denver, United States; [2]Department of Biological Sciences, University of Denver, Denver, United States

**Abstract** Oriented cell intercalation is an essential developmental process that shapes tissue morphologies through the directional insertion of cells between their neighbors. Previous research has focused on properties of cell–cell *interfaces*, while the function of tricellular *vertices* has remained unaddressed. Here, we identify a highly novel mechanism in which vertices demonstrate independent sliding behaviors along cell peripheries to produce the topological deformations responsible for intercalation. Through systematic analysis, we find that the motion of vertices connected by contracting interfaces is not physically coupled, but instead possess strong radial coupling. E-cadherin and Myosin II exist in previously unstudied populations at cell vertices and undergo oscillatory cycles of accumulation and dispersion that are coordinated with changes in cell area. Additionally, peak enrichment of vertex E-cadherin/Myosin II coincides with interface length stabilization. Our results suggest a model in which asymmetric radial force balance directs the progressive, ratcheted motion of individual vertices to drive intercalation.
DOI: https://doi.org/10.7554/eLife.34586.001

*For correspondence:
jblanke4@du.edu (JTB);
Dinah.Loerke@du.edu (DL)

†These authors contributed equally to this work

Competing interests: The authors declare that no competing interests exist.

## Introduction

A common characteristic of many tissues and organisms is an elongation along a primary dimensional axis. The oriented intercalation of cells is one of the fundamental mechanisms utilized to direct tissue elongation (*Keller et al., 2000*). Tissue elongation is essential to the shaping of an elongated body axis (*Keller et al., 2000*; *Irvine and Wieschaus, 1994*), as well as the development of many internal organs, such as the palate, cochlea, gut, and kidney (*Chalmers and Slack, 2000*; *Tudela et al., 2002*; *Wang et al., 2005*; *Lienkamp et al., 2012*). Epithelial cell intercalation drives elongation of the *Drosophila* body axis during gastrulation, in a process known as germband extension (GBE; *Irvine and Wieschaus, 1994*). The intercalary behaviors driving GBE occur through a remodeling of cell topologies, with cells contracting shared anterior-posterior (AP, vertical or T1) interfaces to a single point, followed by newly juxtaposed dorsal-ventral (DV) cells constructing horizontally-oriented interfaces between them (*Irvine and Wieschaus, 1994*; *Bertet et al., 2004*; *Blankenship et al., 2006*; *Collinet et al., 2015*; *Yu and Fernandez-Gonzalez, 2016*). This is referred to as a topological T1 process, and results in a cumulative contraction of the embryonic epithelium along the DV axis, which helps to drive a perpendicular elongation along the AP axis.

Previous research into the genetic factors associated with GBE has shown that global polarizing cues from maternal AP patterning are translated into asymmetric protein distributions at the cellular level (*Irvine and Wieschaus, 1994*; *Blankenship et al., 2006*). At AP interfaces, Myosin II forms both

**eLife digest** Cells need to come together to form tissues of different shapes and sizes. Cells can move about in different ways to shape the tissues. For example, a process called cell intercalation is vital for creating elongated structures like the spinal cord and inner ear. In intercalation, a cell slots itself between neighboring cells to lengthen tissues in one direction.

Most of the work to understand cell intercalation has examined the interfaces that form between two neighboring cells. But there are points called vertices where three cells make contact with each other. Vanderleest, Smits et al. have now used microscopy and computational analysis to examine these contact points, known as vertices, in fruit flies.

It was thought that vertices that are connected by a single interface coordinate how they move. However, Vanderleest, Smits et al. now show that these connected vertices move independently of each other. Instead, the movements of unconnected vertices on opposite sides of the cell show coordination.

Vanderleest, Smits et al. also found that two proteins build up at the vertices in the early stages of intercalation. One of these, called E-cadherin, enables cells to stick to each other. The other protein, called Myosin II, helps E-cadherin to localize to the vertices and also enables cells to contract. These results suggest that the vertices help to guide intercalation and changes in cell shape.

Tracking the vertices over time revealed that they slide around the surface of the cells. During this sliding the total length of the interfaces that meet at the vertex remains the same – so as one becomes shorter, neighboring interfaces will become longer. This creates a zipper-like movement of the vertices that tugs the cells into line and suggests a new mechanism by which interconnected cells can change shape. Future work will focus on identifying the molecules that specify these unique vertex behaviors.

DOI: https://doi.org/10.7554/eLife.34586.002

supracellular cables and smaller, transient networks. Protein populations associated with adhesion (E-cadherin, ß-catenin, Bazooka/Par-3) are found enriched at non-contracting interfaces (*Blankenship et al., 2006*). This body of work led to a model in which actomyosin networks mediate higher line tensions along AP interfaces to direct contraction (*Fernandez-Gonzalez et al., 2009*; *Rauzi et al., 2008*). However, these studies have been limited to the molecular and mechanical characteristics of *interfaces* between two cells. The discrete regions where these interfaces overlap, *tricellular vertices*, have never been comprehensively examined.

As a result of the focus on cell-cell interfaces, many studies on force-generation during intercalation have addressed forces oriented along the cell cortex (*Bertet et al., 2004*; *Fernandez-Gonzalez et al., 2009*; *Rauzi et al., 2008*; *Rauzi et al., 2010*; *Kasza et al., 2014*; *Simões et al., 2014*; *Collinet et al., 2015*; *Yu and Fernandez-Gonzalez, 2016*; *Sun et al., 2017*). However, Myosin II populations are highly active and are transiently present in multiple locations in epithelial cells, including in medial and apical cell regions (*Rauzi et al., 2010*; *Fernandez-Gonzalez and Zallen, 2011*; *Sawyer et al., 2011*; *Sun et al., 2017*). These medial actomyosin networks drive a number of morphogenetic processes by mediating oscillations in cell area (*Martin et al., 2009*; *Roh-Johnson et al., 2012*; *Simões et al., 2017*; *An et al., 2017*). Indeed, during GBE medial Myosin II flows direct apical area oscillations that contribute to AP/DV anisotropy within a cell (*Rauzi et al., 2010*; *Fernandez-Gonzalez and Zallen, 2011*; *Sawyer et al., 2011*). However, the mechanisms by which they could be linked to cell-neighbor exchange have been unclear.

Here, we demonstrate that vertices move independently of one another during T1 contraction, and exhibit distinct molecular dynamics that are required for effective intercalation. We show that intercalation proceeds through a sliding vertex mechanism that physically couples vertex motion to radially-oriented forces. E-cadherin and Myosin II are strikingly enriched at vertices, and this vertex enrichment coincides with length stabilization post-sliding. E-cadherin recruitment at vertices is coordinated with apical cell area oscillations, and is favored at vertices associated with AP interfaces. Finally, perturbing Myosin II function reduces E-cadherin enrichment and dynamics at vertices, and leads to a loss of productive intercalation. Together, these observations provide a mechanism by

which area oscillations are coupled to cyclic molecular dynamics, and further introduce a link between the molecular properties of tricellular vertices and the emergent biophysical properties of the tissue.

## Results

### E-cadherin is dynamic and enriched at tricellular vertices

To study in more detail how cell topologies are remodeled, we first examined the localization of endogenously tagged E-cadherin (E-cad:GFP) at the onset of intercalation. Strikingly, and in contrast to the previously reported homogenous distribution of E-cadherin along DV interfaces at mid-GBE (*Blankenship et al., 2006*), we found that E-cadherin was highly enriched at vertices (*Figure 1A, B and C*, and *Figure 1—figure supplement 1A–C*). This strong vertex-association of E-cadherin began at the onset of GBE, with E-cadherin diffusely present at apical cell interfaces prior to intercalation (*Figure 1D and D'*). Systematic quantification of vertex-associated E-cadherin averaged across the entire time of GBE (see Materials and methods; *Figure 1—figure supplement 1D*) also revealed that E-cadherin maintains a higher vertex:interface ratio than does a control, plasma membrane-associated marker (Gap43:mCh; *Figure 1B and C*). Interestingly, although E-cadherin is enriched at vertices, it possesses a much wider distribution of intensities than the control marker, which suggested that vertex associated E-cadherin might undergo cycles of enrichment and dispersion (*Figure 1C*). Analysis of vertex E-cadherin confirmed that the intensity of enrichment fluctuates in time (*Figure 1E and F*; *Video 1*). These results show that the vertex-specific localization of E-cadherin temporally coincides with intercalary movements, and suggest that vertex E-cadherin may function in directing GBE.

### Movement of cell vertices is physically coupled in the radial direction

As previous literature has focused on interface behaviors during GBE, and since we observed predominantly vertex-localized E-cadherin, we next developed a computational method for following individual vertex trajectories during interface contraction. A central physical expectation from previously described line-tension models (*Fernandez-Gonzalez et al., 2009*; *Rauzi et al., 2010*) is that the inward movement of vertices connected by a contracting interface should show evidence of mechanical coupling (*Figure 2A*). Surprisingly, however, we observed no evidence of this hypothesized coupling (*Figure 2B and C*). Indeed, in systematic pair-wise analyses of cell vertices, physical coupling could only be observed in *radial* directions (e.g. between vertices 3 and 6; *Figure 2B' and C*). In other words, an inward correlation of vertex motion was only found between vertex pairs on opposite sides of the cell, with the largest correlations between those diametrically opposed (*Figure 2C*). These results indicate that during the contraction of an AP interface, the motion of the two vertices toward the middle of the interface (referred to as 'productive' motion) occurs independently of each other, while all vertices undergo coupled motion into the radial direction. These results argue against a line tension-driven model of interface contraction, and suggest that intercalary movements should be reconsidered in terms of cell vertices and radially exerted forces.

### Vertices slide independently of one another

As we continued our analysis of vertex steps in vivo, a novel behavior began to emerge: cell vertices associated with contracting interfaces often underwent periods of *productive sliding* along the plasma membrane (*Figure 2D*; *Video 2*). This suggested that the uncoupled motion of T1 vertices is due to vertex sliding, a previously uncharacterized form of cell-shape deformation. Measurement of interface lengths showed that as a vertical interface contracts (*Figure 2D and E*; blue) the interface adjacent to it elongates (*Figure 2D and E*; red), and consequently the total length stays constant (*Figure 2D and E*; black). This compensatory increase in adjacent interface length is contrary to what would be expected through canonical models of interface contraction, in which the contracting interface shortens while adjacent interfaces maintain a constant length. Additionally, analysis of the lengths of all contracting and adjacent interfaces throughout GBE demonstrated that this behavior is a systematic component of T1-associated vertex movements (*Figure 2F*). We also observe instances where vertices take turns sliding over the course of a T1 contraction (*Figure 2G and H*). These results are consistent with the observed, uncoupled movement of vertices, and provide a mechanism

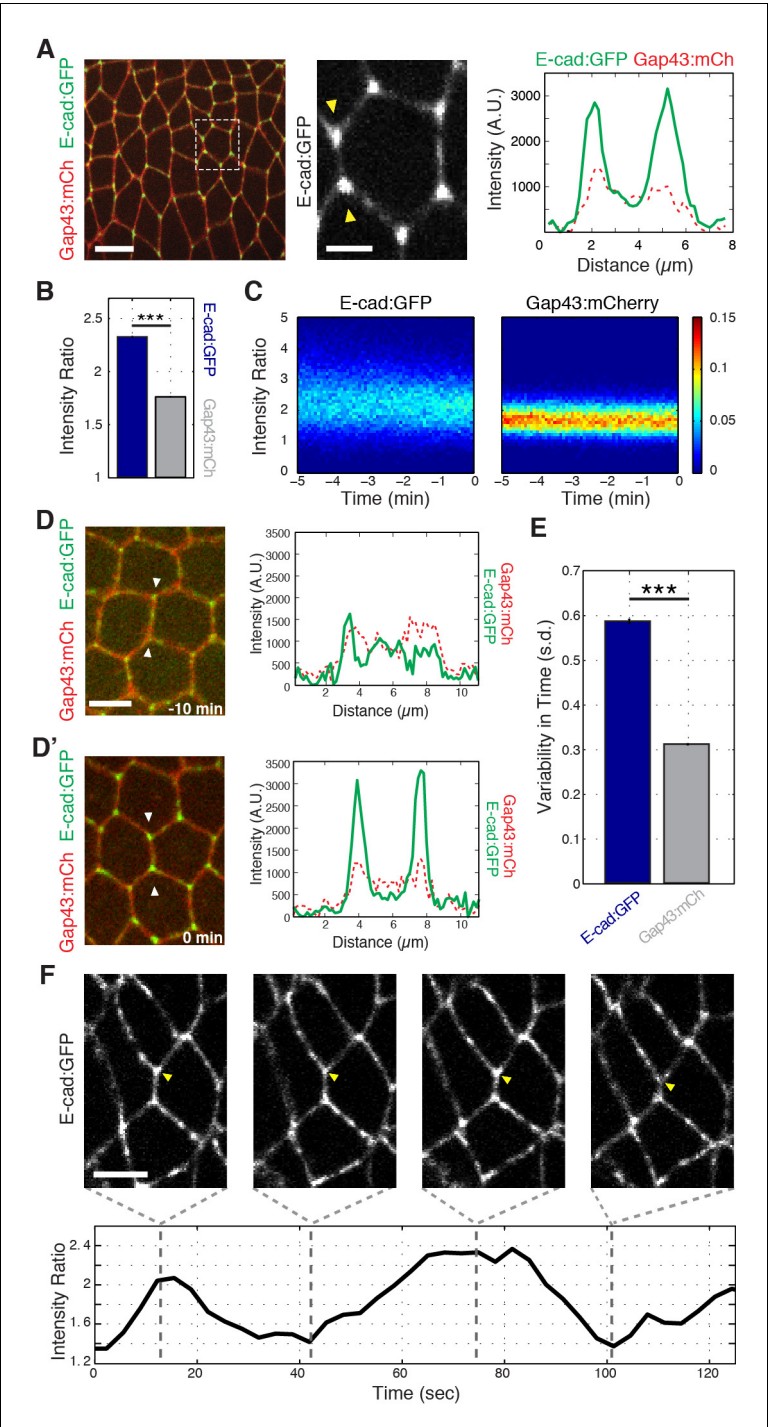

**Figure 1.** Enrichment and dynamics of E-cadherin at vertices. (**A**) Vertex enrichment of E-cadherin at the initiation of GBE. Endogenous locus E-cad:GFP (green) and plasma membrane marker (Gap43:mCh, red). Single pixel intensity line plot (right) for both E-cad:GFP and Gap43:mCh from region between the yellow arrow heads (middle panel). (**B**) Vertex:interface intensity ratios computed for each vertex and frame (n = 87662 vertex time points). (**C**) Heat maps showing the distribution of vertex intensity ratios of Ecad:GFP (left) and Gap43:mCh (right) as a function of time over all embryos aligned to their most contracted state. Color bar shows relative frequency. (**D**–**D'**) T1 configuration of cells expressing E-cad:GFP and membrane marker (Gap43:mCh) imaged 10 min before start of GBE (**D**) and at the onset of intercalation (**D'**). Intensity line plot corresponds to the single pixel intensity line drawn between the two arrowheads in the left panel. (**E**) Vertex-associated E-cad intensities display greater variation than control Gap43:mCh membrane marker. Standard deviation over time for a given vertex's intensity
*Figure 1 continued on next page*

*Figure 1 continued*

ratio averaged (n = 3188 vertex trajectories). (F) Oscillations in E-cad intensity at an individual T1-associated vertex (yellow arrowhead). The intensity ratio is plotted over time (bottom). Scale bars are 10 μm (**A**, left), 3 μm (**A**, middle), and 5 μm (**D, F**). The data are from five embryos and represent mean ± s.e.m. Statistical tests were done by Student's t-test. *** denotes p<0.0001.

DOI: https://doi.org/10.7554/eLife.34586.003

The following figure supplement is available for figure 1:

**Figure supplement 1.** Method of segmentation and vertex analysis showing that vertices become increasingly enriched with E-cadherin prior to the start of cell intercalation.

DOI: https://doi.org/10.7554/eLife.34586.004

by which a single vertex can independently produce the changes in cell shape that drive cell intercalation.

## Interface contraction occurs during periods of cell area contraction

Since our data demonstrate that the motion of vertices is coupled radially, we sought to understand the forces that could drive this movement. Previous research has shown that an apical actomyosin network drives area oscillations during GBE (*Rauzi et al., 2010*; *Fernandez-Gonzalez and Zallen, 2011*); *Figure 3A and A'*; *Video 3*). This appears to be a common feature of many cell shaping processes, as similar apical area oscillations occur in the invagination of the *Drosophila* ventral furrow (*Martin et al., 2009*; *Roh-Johnson et al., 2012*), neuroblast ingression (*Simões et al., 2017*; *An et al., 2017*), and the internalization of the *C. elegans* endodermal precursor cells (*Roh-Johnson et al., 2012*). To study the dynamics of cell vertices and interface lengths in terms of these radially oriented oscillations, we developed a computational assay to identify the instantaneous phase of cell area oscillations. We could then interpolate vertex motion and interface length changes in this area phase space for large numbers of cell oscillations (see Materials and methods; *Figure 3—figure supplement 1A–A'*). These instantaneous phase data follow a coordinate system where area contraction corresponds to angles from −180 to 0 degrees (*Figure 3B and B'*; gray shading), while area expansion occurs in the period from 0 to +180 degrees (*Figure 3B and B'*; blue shading).

Using this method, we measured the systematic changes in interface lengths during apical area oscillations. Under isotropic conditions, the expected theoretical behavior would be that interface lengths would oscillate along with oscillations in cell area (*Figure 3C and C'*; black). However, an intriguing behavior is observed when empirical interface lengths are plotted against cell phase. We observed a larger than isotropic *decrease* in vertical interface length during area contraction (*Figure 3C and C'*; blue; an individual example is shown in *Figure 3—figure supplement 1B*). Notably, this decrease is *preserved* even as area contractions are reversed. Conversely, transverse interfaces undergo a smaller than isotropic decrease in

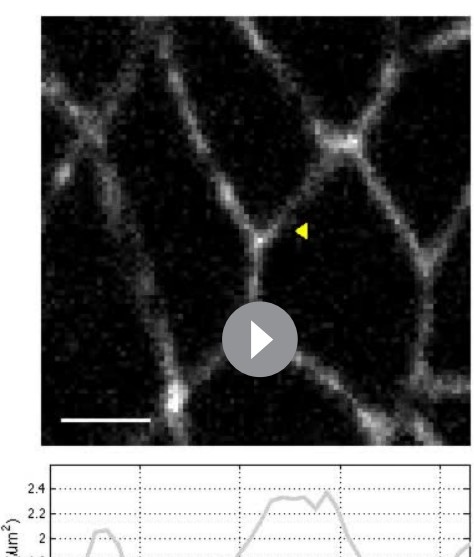

**Video 1.** E-cadherin intensity ratio of vertices are dynamic. Time-lapse images of an embryo expressing the adhesion marker E-cad:GFP during germband extension. The vertex (marked by yellow arrow) exhibits dynamic changes in E-cadherin enrichment. The intensity ratio is plotted as a function of time (bottom). Total time is 125 s, 30 frames/second. Scale bar is three microns.

DOI: https://doi.org/10.7554/eLife.34586.005

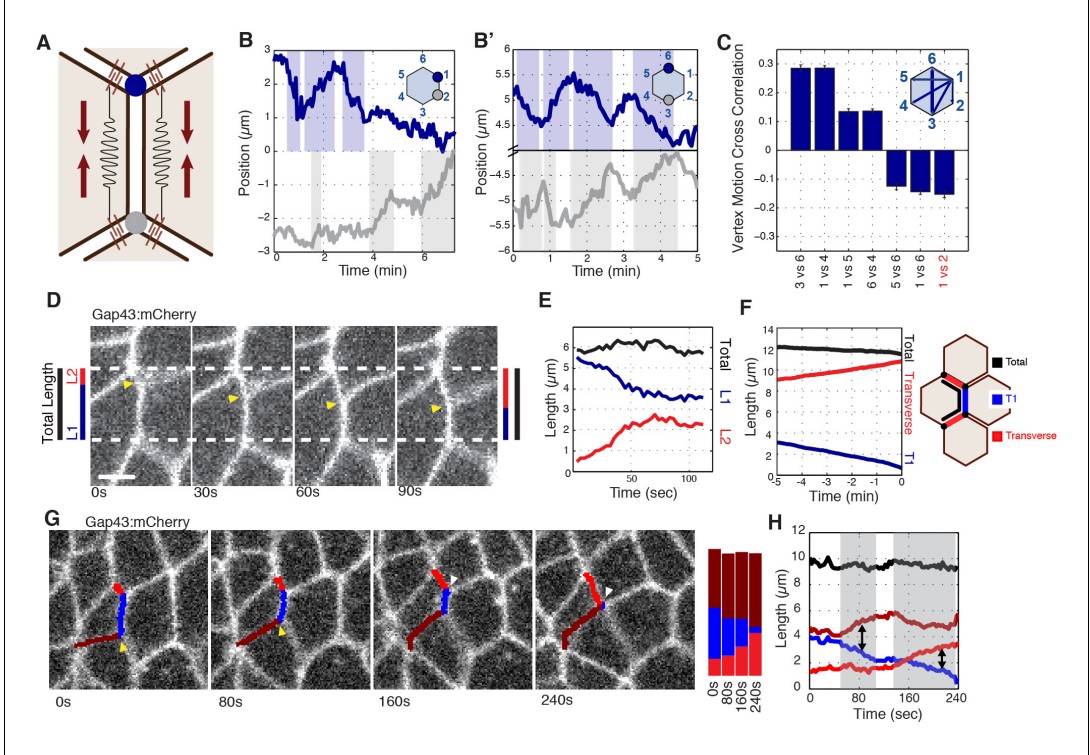

**Figure 2.** Radial coupling and sliding of cell vertices during intercalation. (**A**) Schematic showing line tension model, in which tensioned springs pull across interface lengths on either side of a contracting interface. Blue and gray dots indicate tricellular vertices. (**B**) Vertices at either end of a T1 interface display uncoordinated movements and a lack of physical coupling. Vertex displacement plotted over time. (**B'**) Radial coupling of cell vertices. Vertices that are radially opposed display coordinated movements and coupling of physical displacements. Shaded regions were manually drawn to point out active motion. (**C**) Quantification of cross-correlation between vertex pairs (n = 385, 772, 769, 1551, 716, 824, 436 for vertex pair categories from left to right, data from first 20 min of cell intercalation when T1 behaviors occur). $p<0.0001$ for all vertex pairs. Mean ± s.e.m is shown and one sample Student's t-test was performed with hypothesized mean of 0. (**D**) Total interface lengths (black bar) are conserved during a vertex sliding event, while the contracting L1 interface shrinks (blue). The associated L2 interface (red bar) has a compensatory increase in length as the AP interface contracts (L1, blue bar). Yellow arrowhead shows sliding vertex, white dashed lines mark total length. Scale bar is 5 μm. (**E**) Total, L1, and L2 lengths plotted over time. (**F**) Systematic measurement of all fully contracting interface lengths (n = 168 triplet interfaces). Contracting interfaces are aligned and averaged such that their last time point is set to $t = 0$ (blue curve). The summed lengths of both associated transverse junctions is in red, and the total summed lengths of all three junctions is in black. The data are from five embryos. (**G**) Time series shows a fully contracting T1 interface (blue), and top (red) and bottom (maroon) transverse junctions lengthening over time. The bottom vertex slides first (yellow arrowhead) followed by the top vertex (white arrowhead). (**H**) Plot of the interface lengths (same colors in G) and the sum of the three lengths in black; grey shading and black arrowheads point to sliding events.

DOI: https://doi.org/10.7554/eLife.34586.006

interface length, and appear to undergo a compensatory increase in interface length during area expansion (*Figure 3C and C'*; red). To account for the fact that area contraction and expansion generally are associated with decreases and increases in cell perimeter, we also plotted the effective length changes, or fractional length (defined as interface length divided by cell perimeter), of vertical and transverse interfaces (*Figure 3D*). The fractional length metric has the advantage that isotropic area changes do not result in changes of total fractional length. These results show that about 64% of the effective length contraction of vertical interfaces occurs during area contraction, while only 36% occurs during area expansion when absolute length is stabilized (*Figure 3D*). Thus, vertex sliding occurs opportunistically during area contraction phases in a ratchet-like fashion, so that shortened vertical interface lengths are stabilized during area expansion.

We also used this phasic analysis to examine cell contours during periods when cells are actively contracting in cell area as compared to periods when cells are expanding (*Figure 3E,F*). We generated a convexity/concavity cell shape metric by comparing the area ratio between a theoretical straight-line Euclidean geometry that connects cell vertices to the experimentally defined cell

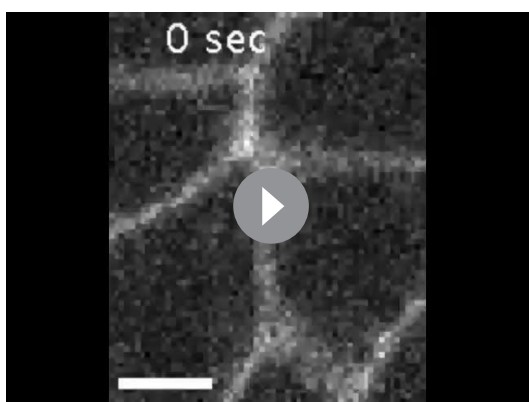

**Video 2.** Contracting vertices slide along cell interfaces. Time-lapse images of an embryo expressing the membrane marker Gap43:mCherry during germband extension. A single vertex can move independently of adjacent vertices and interfaces to result in the cell-shape changes associated with T1 contraction. Anterior is left, ventral is down. Total time is 118.8 s, 25 frames/second. Scale bar is three microns.
DOI: https://doi.org/10.7554/eLife.34586.007

contours. Interestingly, cells possess concave cell contours during contraction, and convex boundaries during apical area expansion (*Figure 3E–H*). These results are consistent with cell vertices, but not cell interfaces, leading overall changes in cell shape and again suggest that cell vertices are key structures that govern cell topologies.

## E-cadherin intensities are in phase with area oscillations and peak with vertex stabilization

We then examined the behaviors of E-cadherin in cells during observed oscillations in apical area. As cell areas contracted, we found specific vertices where E-cadherin became distinctly more enriched (*Figure 4A*, yellow arrowhead). In these instances, the associated T1 interface length decrease observed during area contraction was stabilized during subsequent area expansion. In contrast, other vertices had very little E-cadherin enrichment, and the interface length change of these vertices scaled with area oscillations (*Figure 4A*, white arrowhead). A cross-correlation function relating E-cadherin intensity and cell area showed that as areas decreased, vertex-associated E-cadherin intensity systematically increased (*Figure 4B*; *Figure 4—figure supplement 1A–A'''*). Interestingly, this cross-correlation also showed that vertex E-cadherin intensity peaks just before cell area is in its most contracted state and before the onset of area expansion (*Figure 4B*). Together, this phase relationship suggests that E-cadherin intensity is coordinated with cell area oscillations, and that enrichment of E-cadherin acts to stabilize sliding vertices. These results further indicate a model in which area oscillations provide the force for intercalation, with E-cadherin dynamics functioning as a molecular ratchet to harness vertex-sliding into productive movement.

We then hypothesized that if oscillations in E-cadherin enrichments are essential to vertex movements and sliding, then artificially stabilizing and/or increasing E-cadherin at the plasma membrane should disrupt vertex displacement. We therefore inhibited endocytosis by injecting embryos with a small-molecule inhibitor, chlorpromazine (*Levayer et al., 2011*). This increased the total amount of E-cadherin at cell vertices and interfaces compared to control embryos; additionally, cells maintained a vertex-associated enrichment of E-cadherin throughout GBE (*Figure 4C and D*; *Figure 4—figure supplement 2A*). Importantly, endocytic inhibition prevented E-cadherin from oscillating to a lower enrichment state (*Figure 4E and F*; *Figure 4—figure supplement 2A*; *Video 4*). Although E-cadherin dynamics were disrupted, oscillations in cell areas still occurred (*Figure 4—figure supplement 2B*; *Video 4*). However, under these conditions we found that there was virtually no difference between the theoretical isotropic length change and the observed length changes for either vertical or transverse interfaces (*Figure 4G*). To account for the possibility that vertex sliding is reduced due to other effects of endocytic inhibition we also measured length changes in E-cadherin overexpressing embryos and found that length ratcheting is also reduced (*Figure 4—figure supplement 2C–C''*). Indeed, vertices in Ubiquitin-E-cadherin:GFP overexpressing embryos had displacements that were reduced by 41% as compared to control embryos (*Figure 4—figure supplement 2C''*). Thus, stabilizing E-cadherin prevents vertex sliding and uncouples the ratcheted motion of individual vertices from the motive oscillations in cell area. This further suggests that increased E-cadherin levels restrict vertex sliding, and supports a model in which E-cadherin enrichment at the end of an apical area oscillation serves as a ratchet that stabilizes vertices post-sliding.

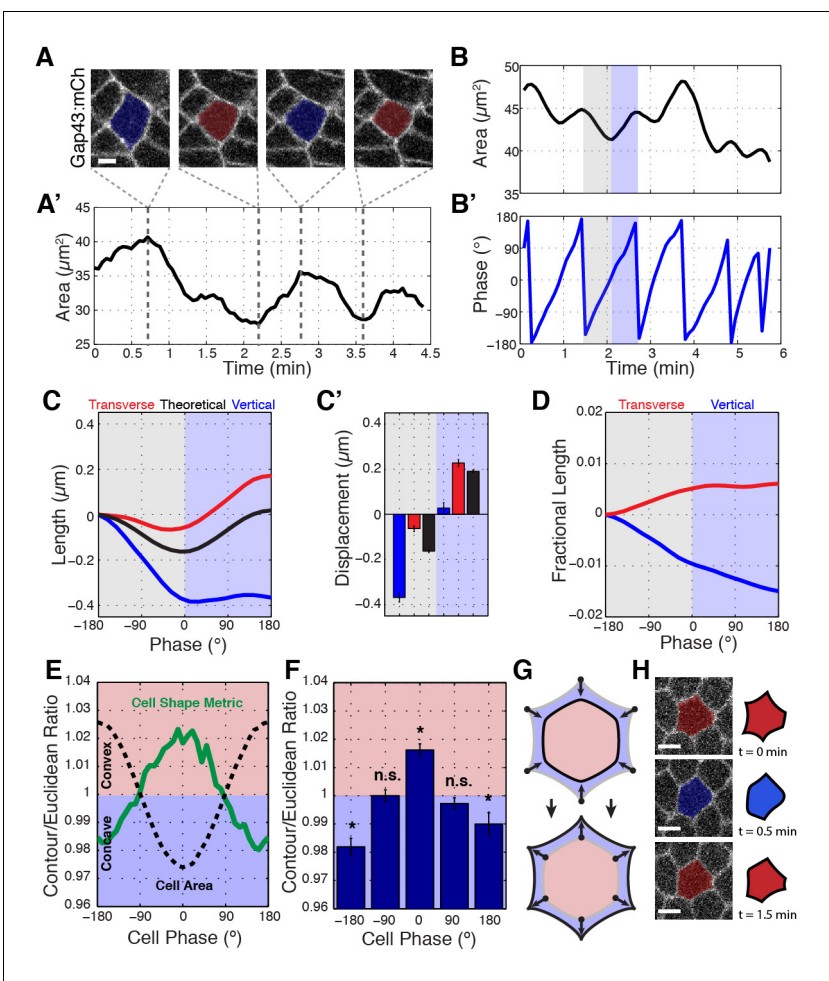

**Figure 3.** Ratcheted vertex movement is coordinated with changes in apical cell area. (**A–A'**) Intercalary cell undergoing apical area oscillations colored blue for maxima and red for minima (**A**) and plot of the area over time (**A'**). Gray dashed lines indicate image time points. (**B**) Plot of cell area trace over time. An oscillatory cycle is highlighted with gray and blue for the decreasing and increasing phases, respectively. (**B'**) Phasic plot of the cell in (**B**) with the same highlighted regions shown. (**C**) Vertical (blue), transverse (red), and theoretical (black) interface length change interpolated into phase space of the associated cell's area oscillations, n = 212 vertical interfaces. (**C'**) Quantification of the total length change per decreasing (gray side) and increasing (blue side) half-cycles. (**D**) Fractional length (Length/Perimeter) change for the same interfaces as analyzed in (**C, C'**). Vertical interfaces show contraction in fractional length, while transverse interfaces possess compensatory increases in cell length, consistent with vertex sliding. (**E**) Mean contour over Euclidean cell area ratio (or cell shape metric) verses cell area phase (green curve) and cell area (for reference, black dashed curve), n = 304 cells. (**F**) Quantification of the cell shape metric at specific phase bins. (**G**) Illustration showing that cells at area maxima are more concave, while cells at area minima bulge outwards. (**H**) Sample raw (left) and cartoon (right) cell during one area oscillation. Scale bars are 3 μm (**A**) and 5 μm (**H**) and data are from five embryos, the first 20 min of cell intercalation. The data shown in (**C', F**) represent mean ±s.e.m. (**F**) One sample Student's t-test was performed with hypothesized mean of 1. * denotes p<0.01.

DOI: https://doi.org/10.7554/eLife.34586.008

The following figure supplement is available for figure 3:

**Figure supplement 1.** Instantaneous phase of area oscillations and junction ratcheting.

DOI: https://doi.org/10.7554/eLife.34586.009

## Cell-specific phase anisotropy drives vertex displacements

To this point, our phase-based area analysis had focused only on individual cell behaviors (*Figure 5A*). However, since vertices receive molecular and mechanical inputs from three different

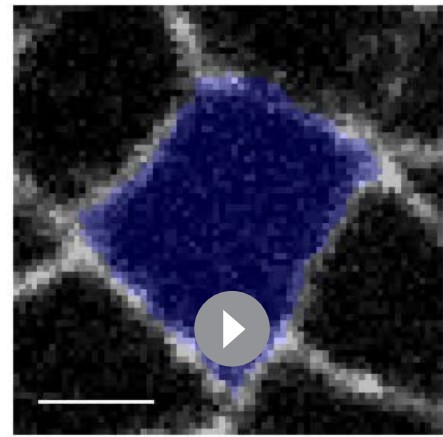

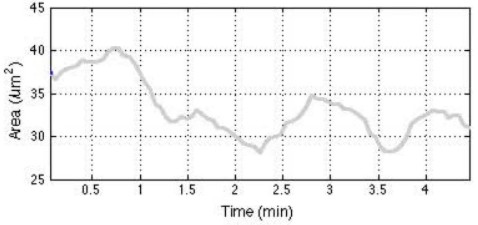

**Video 3.** Cell area oscillations during germband extension. Time-lapse images of a cell (shaded blue) expressing the membrane marker Gap43:mCherry undergoing area oscillations (top) and a plot of the cell area (bottom) over time. Total time is 264 s, 45 frames/second. Scale bar is three microns.

DOI: https://doi.org/10.7554/eLife.34586.010

cells (*Figure 5B*), we developed new computational approaches to study vertex motion with respect to the phases of all three involved cells. To do so, we first decomposed the displacement of a given vertex into two independent components: one tangential and one radial to the cell center (*Figure 5A'*, cyan and blue lines, respectively). Since radial displacement would be expected from isotropic area contraction, productive sliding displacements were defined from the analysis of tangential motions (*Figure 5A'*, cyan). This was then plotted against the phases of each cell that share a common vertex, permitting the measurement of the relative contribution of each cell to productive vertex motion. When tangential displacement rates were plotted with the phases of cells A and B (the two AP neighboring cells at a T1 interface, *Figure 5C'*), productive displacements occurred when both cells were contracting (*Figure 5C, C', E and E'*). However, when the phases of cells A and C were examined, productive displacements only occurred during the coordinated *expansion* of cell C with the *contraction* of cell A (*Figure 5D, D', E and E'*). Strikingly, the magnitude of displacement was much greater in this latter condition than in the first condition (*Figure 5E and E'*). This demonstrates that the phase of cell C is more predictive of productive vertex sliding events than cells A or B, and provides a mechanism for how anisotropic, radial force balance between three cells sharing a common vertex could couple individual T1-associated vertex steps to cell area oscillations.

In light of this phase-correlated mechanical anisotropy, we then asked whether the molecular properties of vertices also displayed symmetry-breaking differences between AP and DV interface-associated vertices. We therefore examined E-cadherin intensities with respect to both area phase and vertex position (*Figure 5F*). As before, we observed a strong increase in E-cadherin intensity as cell areas contracted (*Figure 5G*). Interestingly, this correlation was observed specifically in vertices with polar angles near 90°, corresponding to vertices associated with T1 interfaces (*Figure 5G and H*). Indeed, only contractile, T1-associated vertices show the maximal intensity increases that peak at the end of area contractions, while DV-associated vertices experienced reduced E-cadherin recruitment (*Figure 5G and H*). Stabilizing E-cadherin by disrupting endocytosis reduced the correlation between area phase and vertex movements (*Figure 5—figure supplement 1*), again consistent with E-cadherin dynamics directing productive vertex sliding events. These results demonstrate that E-cadherin is specifically recruited to contracting vertices during apical cell area oscillations and suggest a mechanism to achieve anisotropic stabilization of vertex movements.

## Myosin II directs phasic E-cadherin enrichment at cell vertices

Previous work has shown that medial Myosin II networks flow towards cell interfaces, and locally cluster E-cadherin for endocytic uptake (*Rauzi et al., 2010*; *Levayer et al., 2011*). This led us to examine whether medial Myosin II networks could similarly influence the localization and dynamics of E-cadherin at vertices. Simultaneous imaging of Myosin II:mCherry and E-cad:GFP revealed that, in addition to its previously described junctional and medial populations, Myosin II is highly, and dynamically, enriched at vertices (*Figure 6A–A''*; *Figure 6—figure supplement 1A,B–B''* with Zipper:GFP). This unexpected population of Myosin II exhibited a strong colocalization with vertex-associated E-cadherin (*Figure 6A and A'*). Furthermore, systematic quantitation of vertex Myosin II and

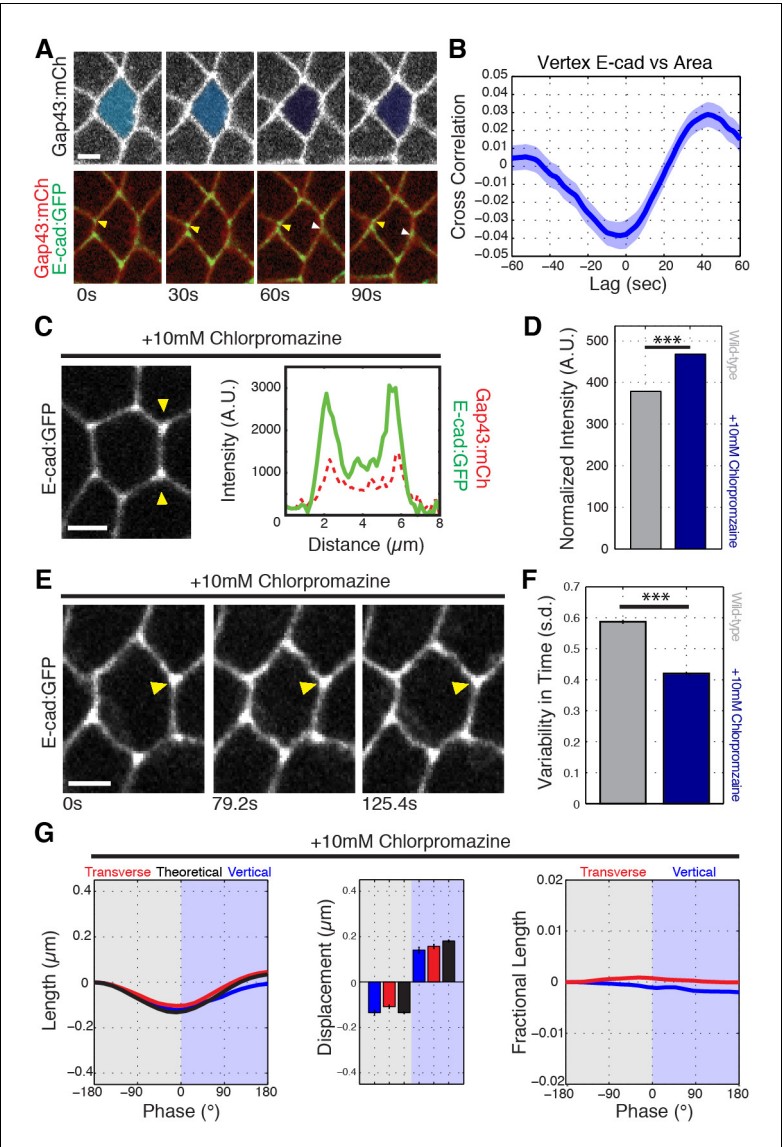

**Figure 4.** E-cadherin stabilizes cell vertices. (**A**) Example of a cell contracting in area while vertical interfaces contract. Top: Gap43:mCh (plasma membrane) channel with cell color-coded such that darker blue represents smaller apical area. Bottom: yellow arrowhead points to vertex E-cad:GFP that increases as the area contracts and interface length is stabilized at shortened length. White arrowhead shows vertex with less E-cad, where interface length increases during expansion phase following contraction. (**B**) Cross-correlation of vertex E-cad intensity and cell area rates of change (n = 610 cells). (**C**) Stabilization of E-cadherin by chlorpromazine injection. Single pixel intensity line plot between the yellow arrowheads shows vertices maintain enrichment of E-cad. (**D**) Quantification of normalized vertex intensities in control and chlorpromazine-injected embryos for E-cad:GFP (blue) and Gap43: mCh (red) over all vertices and time points in the last 5 min before the most contracted state (control: 87662 vertex time points; chlorpromazine: 60935 vertex time points. (**E**) Time series of chlorpromazine-injected E-cad:GFP embryo shows vertex (yellow arrowhead) that does not fluctuate in intensity. (**F**) Averaged standard deviation over time for each vertex's intensity ratio (n = 3188 control and n = 1934 chlorpromazine vertex trajectories). (**G**) Length analyses for chlorpromazine injected embryos (n = 219 vertical junctions). Data are from five embryos (control) and three embryos (chlorpromazine) and the first 20 min of cell intercalation. Scale bars are 3 μm. The data shown in (**B, D, F, G**) represent mean ± s.e.m. Statistical tests were done by Student's t-test. *** denotes p<0.0001.
DOI: https://doi.org/10.7554/eLife.34586.011

The following figure supplements are available for figure 4:

**Figure supplement 1.** Vertex E-cadherin enrichment is anti-correlated with area oscillations.
DOI: https://doi.org/10.7554/eLife.34586.012

*Figure 4 continued on next page*

*Figure 4 continued*

**Figure supplement 2.** Stabilized E-cadherin displays a reduced dynamic range while still possessing oscillations in apical cell area, but junction ratcheting is reduced.
DOI: https://doi.org/10.7554/eLife.34586.013

vertex E-cadherin demonstrated a strong cross-correlation, showing that the dynamics of enrichment of both proteins are temporally coupled (*Figure 6B*, *Figure 6—figure supplement 1C–C′′′, D*). Interestingly, this cross-correlation function also revealed a slight temporal offset, with Myosin II becoming enriched at vertices about 3.3 s before E-cadherin (*Figure 6B*). Similar to E-cadherin (*Figure 5G*), phase-based analysis of Myosin II indicated an enrichment during periods of cell area contraction that is polarized to T1 associated vertices (*Figure 6C and C′*; *Figure 6—figure supplement 1E* with Zipper:GFP). However, these dynamics again subtly preceded those of E-cadherin (compare *Figure 6C and C′* and *Figure 5G and H*). As was previously described for interfaces (*Rauzi et al., 2010*), live imaging of individual cell behaviors showed that approximately 50% of medial Myosin II flows moved toward vertices, resulting in vertex enrichment (*Figure 6D and E*, and *Video 5*). Taken together, these observations show that Myosin II dynamics at vertices are similar to, but temporally precede, those of E-cadherin, and suggest a role for Myosin II in clustering E-cadherin at vertices. This additionally provides a potential mechanistic link between apical cell area oscillations and fluctuations in E-cadherin intensities at cell vertices.

To investigate if Myosin II is required for vertex-associated E-cadherin behaviors, we functionally disrupted Myosin II by injection of the Rok inhibitor Y-27632. Injection of a high concentration of Y-27632 (100 mM) disrupted E-cadherin enrichment to cell vertices, as well as E-cadherin dynamics (*Figures 7A, C and E*). Additionally, analysis of $sqh^{AX}$ mutant embryos produced a similar defect in vertex-associated E-cadherin (*Figure 7—figure supplement 1A–B*). The phenotype of Y-27632 injected embryos was dose dependent, as injection of a lower concentration (25 mM Y-27632) preserved an enrichment of E-cadherin at cell vertices (*Figure 7B and D*). In this background, however, E-cadherin was not significantly polarized to T1 associated vertices, and it lacked cycles of vertex enrichment (*Figures 7D, F and G*). As a consequence, vertex displacement was severely disrupted in Y-27632 injected embryos at both concentrations (*Figure 7H and I*). Interestingly, although productive vertex displacements are observed (*Figure 7H and I*, red areas), these displacements are offset by backwards displacements (*Figure 7H and I*, blue areas). These results are consistent with Myosin II functioning upstream of the ratchet-like behavior of E-cadherin at vertices. It is additionally important to note that cell area oscillations are highly disrupted in Y-27632 injected embryos (*Figure 7—figure supplement 1C*), and that the observed lack of E-cadherin dynamics could represent a disruption of coupling between vertex populations of E-cadherin and medial actomyosin-dependent area oscillations. Either way, these results are consistent with a functional role for Myosin II in clustering E-cadherin at vertices and directing the stabilization of productive sliding movements.

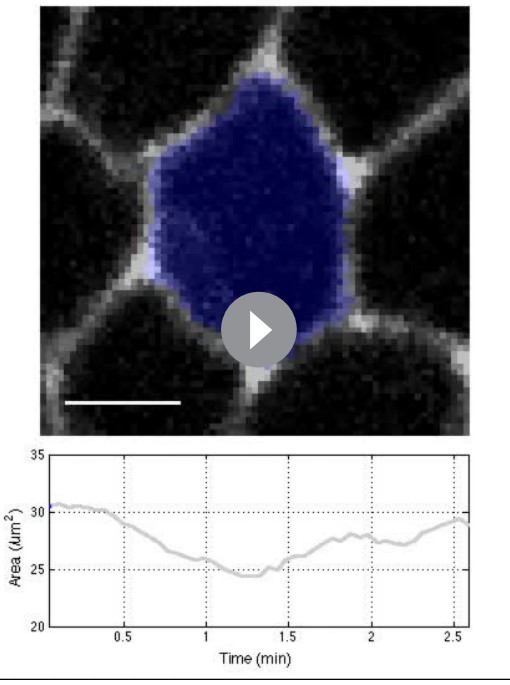

**Video 4.** Cell area oscillations in chlorpromazine injected embryo. Time-lapse images of a cell (shaded blue) expressing the adhesion marker E-cad:GFP undergoing area oscillations (top) and a plot of the cell area (bottom) over time. Total time is 151.8 s, 45 frames/second. Scale bar is three microns.
DOI: https://doi.org/10.7554/eLife.34586.014

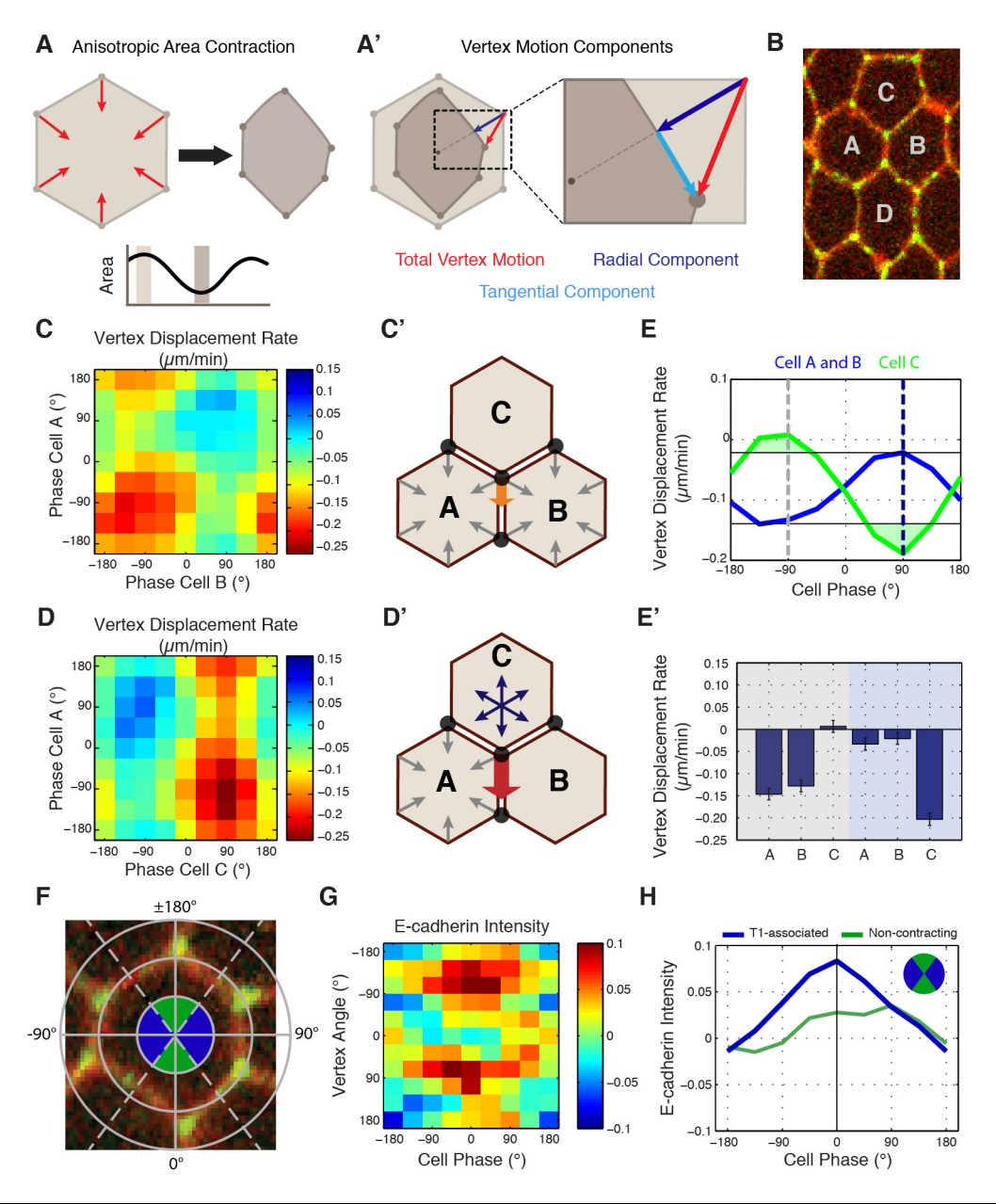

**Figure 5.** Cell-specific phase anisotropy drives vertex displacements. (**A**) Illustration of cell undergoing anisotropic apical area contraction. Red arrows represent inward contractile force. (**A'**) Illustration of vertex displacement (red arrow) broken into tangential (cyan) and radial (dark blue) motion. (**B**) Cell labeling scheme: cells A and B share a common vertical, T1 junction, while cells C and D are to the top and bottom of T1-associated vertices, respectively. (**C, D**) Heat map of the tangential vertex displacement rate with respect to the area phases of cells A and B (**C**) and cells A and C (**D**), (n = 171 T1 interfaces from five embryos). (**C', D'**) Model schematics showing net displacement of a vertex (orange arrow) when cells A and B are contracting in phase (**C'**), and a greater net displacement if A and C are in opposite phases with A contracting and C relaxing (**D'**). (**E**) Plot of the average rate of tangential vertex displacement for cells A and B (blue) and cell C (green), the black horizontal lines show the max and min levels of cells A and B. (**E'**) Quantitation of vertex displacement rates at peak contraction and peak expansion (phases ± 90 degrees). Gray and blue shading corresponds to gray and blue dashed lines in (**E**). (**F**) Illustration of vertex angle assignment used in (**G, H**). Ventral direction is assigned as angle zero. Dashed lines separate regions associated with T1 vertices (blue) and non-contracting vertices (green) angle categories. (**G**) Heat map of the normalized E-cad:GFP intensity of vertices with respect to the angle of the vertex and the cell area

*Figure 5 continued on next page*

*Figure 5 continued*
phase (n = 238 cells from three embryos). (**H**) Plot of average normalized E-cad:GFP intensity for T1-associated and non-contracting vertices with respect to area phase. Data are from first 20 min of cell intercalation.
DOI: https://doi.org/10.7554/eLife.34586.015
The following figure supplement is available for figure 5:

**Figure supplement 1.** Reduced vertex displacements in stabilized E-cadherin embryos.
DOI: https://doi.org/10.7554/eLife.34586.016

## Discussion

In summary, we have shown that radial force coupling drives ratchet-like contractions of AP interfaces. These results also introduce a new functional unit capable of regulating cell topologies – tricellular vertices. We identify a new mechanism driving cell shape remodeling, in which tricellular vertices

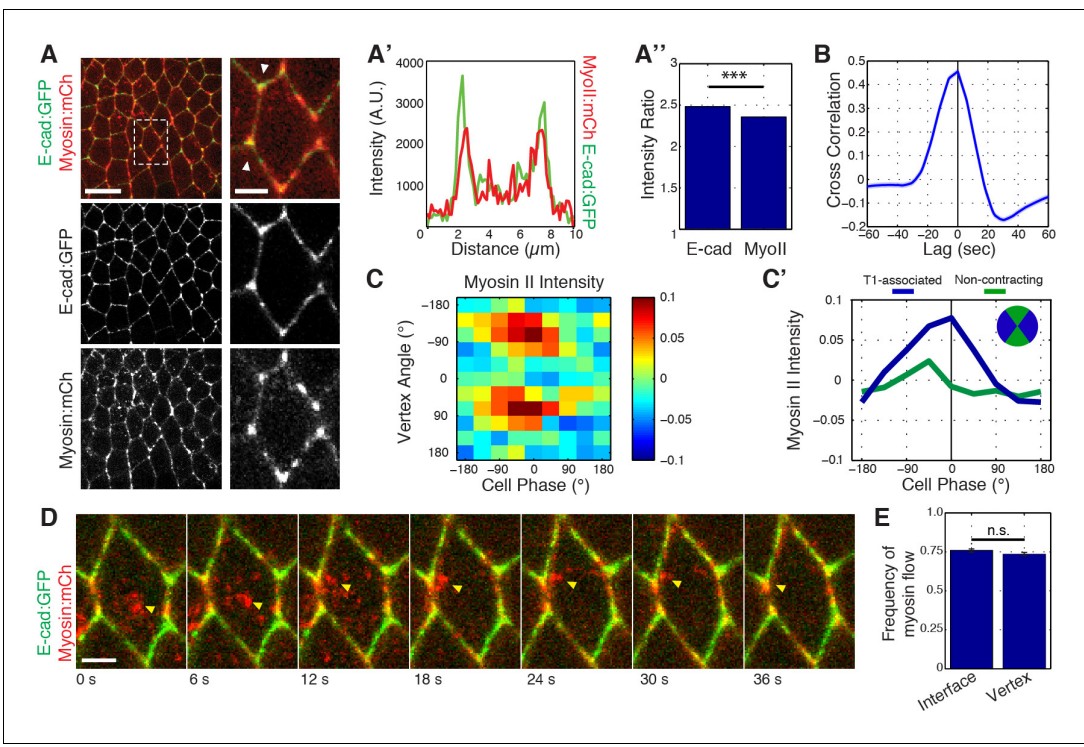

**Figure 6.** Myosin II dynamics at vertices correlate with, but slightly precede, E-cadherin. (**A**) Live imaging of Myosin II (mCh:Sqh, red) and E-cad:GFP (green) embryos during GBE. (**A'**) Single pixel intensity line plot between arrowheads shown in (a). (**A''**) Quantification of Myosin II (mCh:Sqh) and E-cad:GFP vertex intensity ratios (n = 44356 vertex time points). (**B**) Cross-correlation of the rate of change of intensity ratios for E-cad versus Myosin II (n = 800 vertices). (**C**) Heat map of the normalized Myosin II vertex intensity with respect to the vertex angle and area phase, n = 238 cells from three embryos. (**C'**) Plot of average normalized Myosin II intensity for vertical and horizontal vertices with respect to area phase. (**D**) Time sequence showing medial Myosin II flows (yellow arrowheads) toward vertices in merged images. (**E**) Quantification of Myosin II (mCh:Sqh) flow destinations. The frequency shows the number of each myosin flow events per cell per min. (n = 30 cells and 537 Myosin II flows in three embryos, n.s., not significant). Data are from three embryos. Scale bars in (**A**) are 10 μm (left) and 3 μm (right). Scale bar in (**D**) is 3 μm. *** denotes p<0.0001. Statistical tests were done by Student's t-test. (**A, E**) Mean ± s.e.m are shown. Data from first 20 min of cell intercalation.
DOI: https://doi.org/10.7554/eLife.34586.017
The following figure supplement is available for figure 6:

**Figure supplement 1.** Myosin II localizes to cell vertices with E-cadherin, slightly precedes E-cadherin at the vertices, and is preferentially recruited to vertices associated with AP interfaces.
DOI: https://doi.org/10.7554/eLife.34586.018

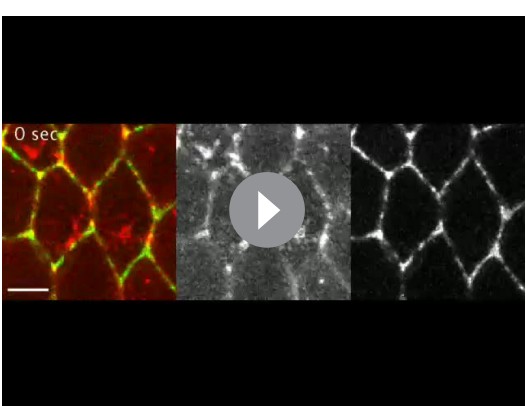

**Video 5.** Myosin II flows toward vertices. Time-lapse images of an embryo expressing E-cad:GFP and mCherry:Myosin II. Myosin II from the medial/apical region of the cell flows toward a vertex, resulting in Myosin II vertex enrichment. Anterior is left, ventral is down. Total time is 48.8 s, 10 frames/second. Scale bar is five microns.

DOI: https://doi.org/10.7554/eLife.34586.019

slide laterally in response to medial force generation (*Figure 7J*). Much of the previous focus in studying intercalary behaviors has been on changes in cell adhesion and force generation at cell interfaces (*Bertet et al., 2004*; *Blankenship et al., 2006*; *Fernandez-Gonzalez et al., 2009*; *Rauzi et al., 2008*; *2010*; *Kasza et al., 2014*; *Simões et al., 2014*; *Collinet et al., 2015*; *Munjal et al., 2015*). While higher line tensions at AP interfaces clearly exist and direct distinct aspects of intercalary cell behaviors (such as interface alignment along the DV axis, recoil velocities upon laser ablation, and boundary element behaviors), it will be interesting to further explore the mechanisms regulating tricellular vertex function (*Rauzi et al., 2008*; *Fernandez-Gonzalez et al., 2009*; *Tetley et al., 2016*). As vertices are connected to three (or more) interfaces as well as the radial coupling reported on here, their displacement will rely on the summed total of these local force contributions. Some of the mechanisms may involve the endocytic uptake of plasma membrane and adhesion proteins at interfaces which have been recently described (*Levayer et al., 2011*; *Jewett et al., 2017*). These molecular models of regulated adhesion are not necessarily dependent on line tensions, and could contribute to the biases in lateral vertex displacements in particular tangential directions. Although our data argue against a predominant function of interface-spanning line tensions in directing interface contraction, local regions of either medial- or interface-associated Myosin II are likely to impact vertex displacements as well. Indeed, although E-cadherin and Myosin II are primarily located at cell vertices early in GBE, by mid-GBE interface localization of both significantly strengthen and is consistent with interface as well as radial contributions to vertex displacements. Thus, cell vertices are well positioned to integrate the many different force-generating networks that will ultimately determine changes in cell shape and topology.

Our results also show that intercellular adhesion dynamics are required for vertex movement. Under conditions in which E-cadherin exhibited greater enrichment and/or stability than is present in wild-type embryos, AP and transverse interface lengths oscillated identically, suggesting that cell vertices were unable to slide productively due to increased adhesive stability. Stabilizing E-cadherin was achieved by inhibiting endocytosis (*Levayer et al., 2011*), and suggests that endocytic events may underlie the dispersion phase of E-cadherin dynamics at cell vertices. Given that endocytic pathways centered on asymmetric planar behaviors of Clathrin, Dynamin, and Rab35-dependent functions have been previously identified (*Levayer et al., 2011*; *Jewett et al., 2017*), it will be interesting to further explore if these same pathways function at or near cell vertices to direct E-cadherin dynamics, and whether endocytosis at vertices or at interfaces is more responsible for vertex E-cadherin redistribution. However, one interesting implication of our work is that these endocytic pathways should alter the balance of adhesion on either side of a vertex, potentially through the uptake of E-cadherin adhesion molecules, but the combined length of T1 and transverse interfaces would remain largely unchanged. This again suggests that the positioning of cell vertices will reflect the combined activities of contractile and adhesion elements that are located directly at the vertex as well as in local regions near the vertex. It also underlines the importance of cell vertices, and suggests a primary importance of vertices in determining cell topologies, which is further indicated by the strong enrichments of E-cadherin and Myosin II at cell vertices during early GBE. Regardless, it will require additional work to tease apart the contributions of E-cadherin stabilization and turnover pathways at cell vertices versus cell interfaces.

We have also shown that as cells contract their apical area, E-cadherin and Myosin II are preferentially recruited to AP vertices to provide the adhesive force necessary to stabilize vertex position and interface length during area relaxation. This, as well as anisotropy in cell area oscillations and local

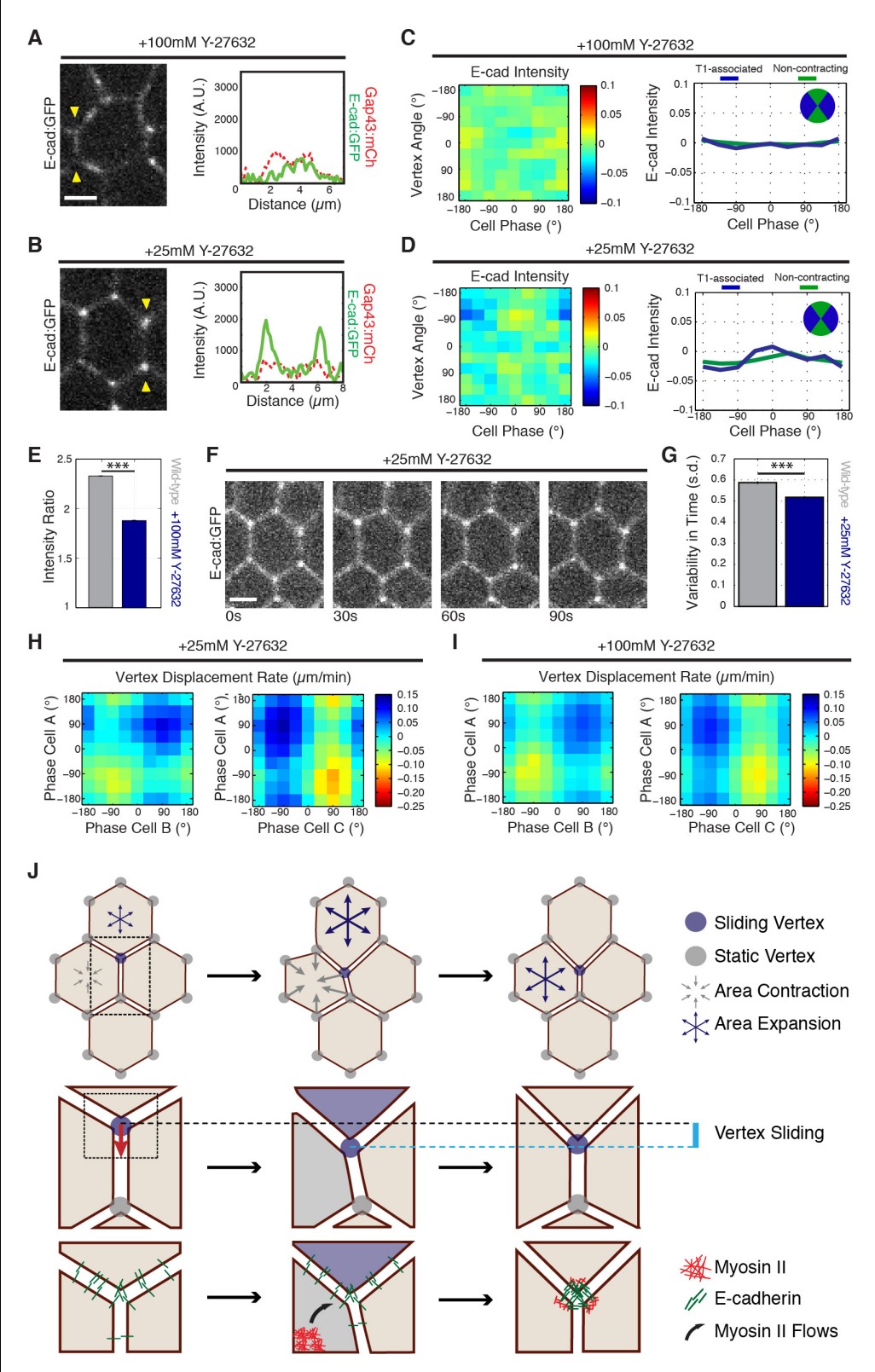

**Figure 7.** Myosin II function is required for E-cadherin dynamics. (**A,B**) E-cad:GFP image of cell and single pixel intensity line plot between arrowheads for 100 mM (**A**) and 25 mM (**B**) Y-27632 injection. (**C,D**) Heat map and plot of the normalized E-cad:GFP intensity at vertices with respect to the angle of the vertex and the area phase for 100 mM (c, n = 409 cells) and 25 mM (d, n = 363 cells) Y-27632 injection. (**E**) Quantification of vertex-to-junction

*Figure 7 continued on next page*

*Figure 7 continued*

intensity ratios measured for each vertex and each frame of 5 min during GBE (n = 86939 vertex time points). (**F**) Time sequence of images showing loss of E-cad:GFP dynamics at vertices in 25 mM Y-27832 injected embryos. (**G**) The averaged standard deviation over time for each vertex's intensity ratio. Control, n = 3188 vertex trajectories; 25 mM, n = 1868 vertex trajectories. (**H, I**) Heat map of tangential vertex motion rates versus the phases of cell A and cell B (left) and cell A and cell C (right) for 25 mM (**H**) and 100 mM (**I**) Y-27632 injection. (**H**) n = 175 junctions. (**I**) n = 220 junctions. (**J**) Model for vertex-directed changes in cell topologies. As a cell adjacent to a vertex along the AP axis contracts (gray arrows), and the adjacent cell along the DV axis expands (blue arrows), the vertex experiences a cumulative asymmetric force, causing it to slide along the interface (middle row, cyan line). At the molecular level, medial Myosin II flows consolidate E-cadherin at cell vertices post-vertex sliding, resulting in a local increase in adhesive stability. This is coordinated with oscillations in apical cell area, which ensures progressive, non-reversible vertex displacements (bottom). Gray shading indicates a contractile phase; blue indicates expansion. Control data from three embryos, Y-27632 25 mM from three embryos, and Y-27632 100 mM from three embryos. Scale bars are 3 μm. *** denotes p<0.0001. (**E, G**) Mean ± s.e.m are shown. Data from first 20 min of cell intercalation.

DOI: https://doi.org/10.7554/eLife.34586.020

The following figure supplement is available for figure 7:

**Figure supplement 1.** E-cadherin vertex enrichment is lost in $sqh^{AX}$ mutant embryos and area oscillation amplitude is reduced under Y-27632 treatment.
DOI: https://doi.org/10.7554/eLife.34586.021

---

imbalances in interface-associated forces (*Rauzi et al., 2010*; *Fernandez-Gonzalez and Zallen, 2011*; *Sawyer et al., 2011*), are likely responsible for enforcing the directionality of intercalation. Our results also suggest that, while radial forces in cells sharing a contracting AP interface are important for vertex displacement, vertex displacement has the strongest correlation with expansive motion in the adjacent, DV-oriented cells. This is intriguing, and suggests a homology to recent results during interface extension in which the adjacent cells provide motive force for extension rather than the cells that share the newly growing interface (*Collinet et al., 2015*; *Yu and Fernandez-Gonzalez, 2016*). At the molecular level, AP vs DV anisotropy at the level of vertices could be a result of stress anisotropy and a mechanosensory feedback loop. Alternatively, vertices may also experience differentially positioned signaling networks, allowing for Myosin II and E-cadherin vertex enrichment.

While the planar sliding behaviors described here are a novel mechanism underlying intercalary behaviors, it is also interesting to note that a similar adherens junction sliding behavior has been observed during three dimensional epithelial folding events as well as during cell ordering in the *Drosophila* notum (*Wang et al., 2012*; *Wang et al., 2013*; *Curran et al., 2017*). In the formation of the epithelial folds that occur on the dorsal surface of the embryo in response to GBE, there is a basal shift in adherens junction position in response to Rap1 signaling events (*Wang et al., 2012*; *Wang et al., 2013*; *Takeda et al., 2018*). It will be intriguing to explore if a similar Rap1-dependent pathway operates on cell vertices during cell intercalation. It is also highly interesting that, in the *Drosophila* pupal notum, a similar conservation of total junctional lengths, referred to as 'continuous neighbor exchange' has been observed (*Curran et al., 2017*). It may well be that the repositioning of junctional/vertex elements will represent a new and conserved paradigm in how cell topologies are re-shaped during development.

Finally, there is a growing body of work on tricellular vertices as unique epithelial structures. Previous studies have shown that cell vertices possess important molecular characteristics capable of coordinating complex cell morphologies (*Staehelin, 1973*; *Graf et al., 1982*; *Schulte et al., 2003*; *Ikenouchi et al., 2005*; *Blankenship et al., 2006*; *Byri et al., 2015*). Indeed, recent work has shown that tricellular junctions act as key sensors of cell shape that serve as landmarks to orient epithelial cell divisions (*Bosveld et al., 2016*), and intestinal stem cells require tricellular function to maintain appropriate homeostatic levels (*Resnik-Docampo et al., 2017*). This suggests that vertices represent unique domains of the cell surface and that tricellular vertices have a special role as centers of functional signaling and physical networks within epithelial sheets.

## Materials and methods

### Image analysis

Image and data analysis were performed in MATLAB. Cells were segmented using a seeded watershed algorithm (see *Figure 1—figure supplement 1A*) and tracked in time. The 'skeletonized' representation of the tissue directly yields vertex positions, interface contours, lengths and orientation angles, apical cell areas and perimeters, which we store together with cell-cell and vertex-vertex connectivity matrices. Cell areas were measured as the sum of the pixels within the contour of the watershed segmentation lines (multiplied by the pixel area). Interface lengths were calculated as the Euclidian distances between the corresponding vertices.

Line plots were generated as single pixel intensities along an interface line drawn manually in ImageJ, and the minimum value was subtracted from all data points and plotted using MATLAB. Images were cropped and leveled in Adobe Photoshop, and figures were prepared in Adobe Illustrator.

### Vertex intensity ratio measurement and quantification

Intensity measurements of vertices, local interfaces, and local background were automated using the watershed segmentation data to generate ROIs (using distance transforms) that were then used to measure average pixel intensities (see *Figure 1—figure supplement 1B* for image of ROIs). Vertex ROIs were generated by making a binary matrix of the vertex pixel and using the distance transform to identify all pixels within 3 pixels of the vertex (forming a pixelated disk seven pixels in diameter). The local interface and background intensities were measured within a 41-by-41 pixel square centered at the vertex, and local interface and background intensities were used to account for non-uniform illumination and varying junctional protein enrichment. Within the square neighborhood, the interface ROI was acquired using the distance transform to identify all pixels within 3 pixels of the interfaces and removing the vertex ROI. The local background ROI was acquired using a distance transform to identify all pixels at least seven pixels in distance from the segmentation lines: the seven pixel value was chosen to create a three pixel buffer zone between the interface/vertex ROIs and the background. The intensity ratio for each vertex was calculated as (V-B)/(J-B) where V is the vertex, B is the background, and J is the junction (interface) intensity measurements. Quantification of the intensity ratio was performed by averaging over all vertex time-points during GBE and all embryos (*Figures 1B*, *6A''* and *7E*) or resolved in time (*Figure 1C*).

### Vertex intensity ratio variability in time measurements

To measure the dynamics of protein enrichment at vertices, we calculated the standard deviation of the intensity ratio trajectory of each vertex. To quantify this we averaged over all vertices.

### Separation of vertex motion and tangential component

With respect to the embryo or the imaging field of view, vertices undergo primarily two different components of motion: drift (or translational motion) as the germband elongates, and cell-centric motion such as cell shape changes and intercalation movements. Thus, to analyze the motions associated with cell shape changes and intercalation, we removed the drift component by measuring vertex positions with respect to the centroid of the cell, $\vec{c}(t) = [x_c(t), y_c(t)]$, where $x_c(t)$ and $y_c(t)$ are the x- and y-coordinates of the cell centroid at time $t$. The centroids of each cell were obtained over time using MATLAB's *regionprops* function. In this cell-centric reference frame the position of a vertex is given by $\vec{v}(t) = [x_v(t) - x_c(t), y_v(t) - y_c(t)]$, where $[x_v(t), y_v(t)]$ is the position of the vertex in the image frame of reference. For measuring vertex tangential motion, vertex positions were converted to polar coordinates, $\vec{r}(t) = [r(t), \theta(t)]$, where $r(t)$ is the radial position of the vertex from the centroid of the cell and $\theta(t)$ is the angle. Tangential displacements of the vertex are given by $\Delta s = r \times \Delta\theta$, where, by convention, displacements towards the opposing vertex of the interface were assigned a negative sign. Thus, interface-contracting displacements of an individual vertex are negative and interface-elongating displacements are positive.

## Correlating vertices coupled direct motion

We quantified the motion coupling of each vertex with each neighbor vertex within the same cell via the cross-correlation at zero-lag of the vertices' rate of displacement towards each other. The motions correlated are the components of displacement in the direction parallel to the two vertices, that is, along the line connecting the pair of vertices. The parallel component is calculated by taking the dot product of the displacement with the average vertex-to-vertex vector before and after displacement, the vertex position trajectories are then computed by taking the cumulative sum of the parallel displacement components. Rates of vertex displacement were calculated over 25 s to avoid correlating localization error (*Weber et al., 2012*). Like pairs of vertices were combined in the quantification (*Figure 2C*): 1 vs. 5 includes 2 vs. 4 representing horizontally coupled vertices, 1 vs. 4 includes 2 vs. 5 representing diagonally coupled, etc. The highlighted plot regions (*Figure 2B and B'*) were manually selected to show periods in which the vertex is highly active in inward or outward motion. Vertex movements towards or away from each other result in a positive correlation, while a negative correlation value indicates that they move in the same direction.

## Vertex sliding shown through sum of contracting and transverse interfaces

One prediction of vertex sliding is length (i.e. plasma membrane) compensation of the adjacent interfaces, that is as a vertex slides one interface gets longer by the same amount that the other gets shorter. To quantify this on time scales of interface contraction, we took the sum of the contracting interface with both of its transverse interfaces (note: this accounts for the sliding motion of both vertices of an interface) to see if total length was conserved. This sum was performed on the last 5 min of all available fully contracting interfaces that had a lifetime of at least 5 min. Of all the fully contracting interfaces that were present at the beginning of the movies this represented 91.8%. Individual cases were aligned such that the last time point was T = 0 min. Each contracting interface has two pairs of two transverse interfaces, those for cell A and for cell B, and both pairs were counted in the average.

## Acquisition of instantaneous phase and amplitude from cell area

To get the instantaneous phase (*Figure 3B'*) of an oscillating signal such as the area of a cell, we used a method known as the Osculating Circle Method (*Hsu et al., 2011*), which is based on the Hilbert Transform but better suited for signals with non-zero mean. To get meaningful results using this method, the signal must be filtered to remove the noise that will result in artificial high frequency oscillations. A Savitzky-Golay filter of polynomial order three and frame size 45 s (the exact number of frames varied based on the frame rate of the movies) was used because it performed best at removing noise and preserving true cell area oscillations (*Figure 3—figure supplement 1A–A'*). Before applying the Savitzky-Golay filter, we de-trended the signal by subtracting a long time scale Gaussian filtered (sigma = 180 s) version of the signal. Instantaneous phase was shifted such that −180 to 0 degrees represents peak-to-trough and 0 to 180 degrees trough-to-peak. The Osculating Circle Method also generates an instantaneous amplitude, which was averaged over time to calculate each cell's average oscillation amplitude (*Figure 4—figure supplement 2B* and *Figure 7—figure supplement 1C*).

## Interpolation of a parameter with respect to phase

Once the instantaneous phase trajectory of a cell's area oscillations is obtained, other time-dependent readouts of the cell dynamics, such as the length of an interface, motion of a vertex, or intensities of a vertex, can be remapped from the time domain to the cell phase domain to average how those parameters change over the course of a cell's oscillation cycle; this averaging is difficult or impossible in the time domain due to variations in oscillation cycle length. Mapping into the phase domain results in a 2D scatter plot of parameter values versus phase values for multiple cycles of phase; we interpolated these data by Gaussian weighted averaging of the parameter to a grid of phase values.

## Interface length and fractional length with respect to area phase

To determine how interface lengths change over the course of an area oscillation cycle the data were mapped into the phase space of the adjacent cells, separately for the phases of cell A and cell B. For each phase cycle the interface length trajectory was shifted (by subtracting the initial length) so that it starts from L = 0 with each cycle. This 'reset' to zero with the start of each cycle was done so that positive lengths represent interface growth and negative lengths represent interface contraction with each phase cycle. The 2D set of data points (phases and shifted lengths) for all phase cycles (from both cell A and cell B, for all AP interfaces, and all movies) were combined and interpolated as described above. For the quantification (*Figure 3C′*) we measured the average shifted length within two 10 degree phase bins: one bin centered at 0 degrees to capture the length change after the area decreasing phase and one centered at 175 degrees (170–180 degrees) to capture the average length change after a full cycle. From the values in the bin centered at 175 degrees we subtract the average of the 0 degree bin to get the length change over just the area expanding phase. The methods for fractional length are the same as above, except that interface lengths are replaced with the ratio of the interface length divided by cell perimeter.

## Contour over Euclidean area ratio

Contour cell area is defined as the area within the watershed segmentation lines plus half the area of the pixels that make up the watershed boundary lines (otherwise the contour areas would be artifactually small). The Euclidean area is defined as the area of the polygon whose vertices are the cell's detected vertices. The contour over Euclidean area ratio (or cell shape metric) is a simple ratio of these two values. When interpolating the cell shape matric to phase space we used the phase of the Euclidean cell area. The quantification in *Figure 3F* was done on 10 degree phase bins centered at ±180, ±90, and 0°.

## Vertex displacement rate with respect to the phases of cells A, B, and C (or D)

To see whether individual vertex motion was correlated to apical cell area oscillation cycles, vertex displacements were tracked with respect to the phases of the three cells that make up the tri-cellular vertex. The particular type of vertex displacement measured was the cell-tangential component, which is perpendicular to the vector from the cell centroid to the vertex, and therefore independent, to the radial motion associated with cell area oscillations. Cell-tangential displacements were assigned a negative sign if the displacement contracted the vertical interface. For each vertex there are two tangential motion trajectories, one with respect to cell A, and one with respect to cell B, thus the methods described here were performed twice for each vertex, once with respect to each cell. For one vertex motion trajectory the instantaneous phases of cells A, B, and the other cell in the triad (C or D) are collected for each time point in the trajectory. Vertex displacements were interpolated to a 3D grid with the three cell phases on the three axes. To account for edge effects of interpolation, data near the edges (±180°) were wrapped around by adding or subtracting 360° to the phases.

## Vertex intensity with respect to cell phase and vertex angle

Vertex intensities of Ecad:GFP and mCherry:Sqh were also interpolated into area phase space in a spatial angle resolved manner. Each vertex has an angle with respect to the centroid of each cell it borders, which also varies over time. These angles were measured with respect to a global reference angle, the angle along the dorsal-ventral axis of the embryo towards the ventral (*Figure 5F*). The intensity trajectory of each vertex was normalized (zero mean and s.d. of 1). Normalized intensities were interpolated to a 2D grid of vertex angle and cell phase angle. To account for edge effects of interpolation, data near the edges (±180°) were wrapped around by adding or subtracting 360° to the phases.

## Cross-correlation of intensity ratios of E-cad and myosin at vertices

To quantify the temporal correlation between E-cad and Myosin enrichment at vertices, we performed a cross-correlation of the rates of change of the vertex intensity ratios (*Figure 6B*). Rates of change were calculated over 25 s to avoid correlating localization error. Signals were normalized

such that mean and standard deviation were 0 and 1, respectively, and an unbiased cross-correlation was computed via MATLAB's *xcorr* function. The temporal lag was estimated by calculating the centroid of the positive peak (*Figure 6B*).

## Measurement of myosin flows to vertices or interfaces

For medial myosin destination measurements, over 160 medial myosin flow events in 10 contracting cells were counted manually in the first 12 min of GBE in each of 3 E-Cad:GFP; mCh:Sqh movies. The flow of myosin to the area of 3 pixels radius around the tricellular vertex was considered as vertex destination, otherwise, it was considered as interface destination.

## Fly stocks and genetics

Stocks were kept at 25°C and maintained by standard procedures. Fly stocks used in this study were endogenous E-cad:GFP (gift of Y. Hong, University of Pittsburg, BL-60584), Zipper:GFP (Kyoto 115–082), Gap43:mCh/TM3 (gift of A. Martin, MIT), Ubi-E-cad:GFP (BL-58471), $sqh^{AX3}$ (BL-25712), and mCherry:Sqh (gift of A. Martin, MIT). E-cad:GFP and Zipper:GFP were expressed from the endogenous loci on the second chromosome and are homozygous viable.

## Live imaging

Embryos were collected on apple juice agar and dechorionated in 50% bleach for 2 min, then rinsed with water and either staged on apple juice agar or transferred to a gas permeable microscope slide and covered with Halocarbon 27 oil. All imaging was performed on a CSU10b Yokogawa spinning disk confocal from Zeiss and Solamere Technologies Group with a 63x/1.4 NA objective, with the exception of Myosin II movies, which were obtained on a CSUX1FW Yokogawa spinning disk confocal from Nikon and Solamere Technologies Group with a 60x/1.4 NA objective. Ecad:GFP; Gap43:mCh images are a summed projection of 5 z-slices taken at 0.5 μm steps starting sub-apically, at ~3 s/frame. Ecad:GFP; mCh:Sqh images are a maximum intensity projection of 8–10 z-slices taken at 0.75 μm steps, at ~6 s/frame.

## Drug injections

Following dechorionation as previously described, embryos were staged and aligned on apple juice agar, glued to a coverslip with heptane glue, and desiccated. Embryos were covered with Halocarbon 700 oil then injected with 10 mM chlorpromazine, or 25 or 100 mM Y-27632. Embryos at the beginning of GBE were injected in the perivitelline space at 50% egg length.

## Repeatability

All measurements were quantified from a minimum of 3 embryos, and represented at least two individual trials.

## Acknowledgements

We are grateful to the generous colleagues who supplied fly lines: E Wieschaus and the Developmental Studies Hybridoma Bank. Thanks are due to the Blankenship and Loerke labs for discussions and critical reading of the manuscript. This work was supported by an NIH R01 GM090065 to JTB, and an NIH R15 GM117463 and Research Corporation for Science Advancement (RCSA) Cottrell Scholar Award 2014 to DL.

## Additional information

### Funding

| Funder | Grant reference number | Author |
| --- | --- | --- |
| National Institute of General Medical Sciences | NIGMS R01 GM090065 | Todd Blankenship |
| National Institute of General Medical Sciences | NIGMS R15 GM126422 | Todd Blankenship |

| Research Corporation for Science Advancement | | Dinah Loerke |
| National Institute of General Medical Sciences | NIH R15 GM117463 | Dinah Loerke |

The funders had no role in study design, data collection and interpretation, or the decision to submit the work for publication.

## Author contributions

Timothy E Vanderleest, Resources, Data curation, Software, Formal analysis, Validation, Investigation, Methodology, Writing—review and editing; Celia M Smits, Conceptualization, Formal analysis, Validation, Investigation, Visualization, Methodology, Writing—original draft, Writing—review and editing; Yi Xie, Cayla E Jewett, Formal analysis, Methodology; J Todd Blankenship, Conceptualization, Formal analysis, Supervision, Funding acquisition, Investigation, Visualization, Writing—original draft, Project administration, Writing—review and editing; Dinah Loerke, Conceptualization, Resources, Software, Formal analysis, Supervision, Funding acquisition, Validation, Investigation, Methodology, Project administration, Writing—review and editing

## Author ORCIDs

J Todd Blankenship (iD) http://orcid.org/0000-0001-8687-9527

## Decision letter and Author response

Decision letter https://doi.org/10.7554/eLife.34586.028
Author response https://doi.org/10.7554/eLife.34586.029

## Additional files

### Supplementary files

• Transparent reporting form
DOI: https://doi.org/10.7554/eLife.34586.022

### Data availability

All data generated or analysed during this study are included in the manuscript and supporting files. Raw images are freely available.

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
