## [Decision Letter]

Thank you for submitting your article "Vertex sliding drives intercalation by radial coupling of adhesion and actomyosin networks" for consideration by *eLife*. Your article has been reviewed by three peer reviewers, and the evaluation has been overseen by a Guest Editor and Didier Stainier as the Senior Editor. The reviewers have opted to remain anonymous.

The reviewers have discussed the reviews with one another and the Reviewing Editor has drafted this decision to help you prepare a revised submission.

Summary:

The manuscript by Vanderleest et al. uses studies of germ band extension (GBE) in *Drosophila* to introduce a new mechanism by which cells can intercalate with one another to lengthen an epithelial tissue. This intercalation process, which is called a T1 transition, has been extensively studied. The current model for this process is that myosin accumulates asymmetrically at vertical cell-cell interfaces and directly shortens them in a ratchet-like manner (line tension model). In this manuscript, the authors perform systematic, quantitative analyses on time-lapse videos of GBE, and then propose an alternate model for how junctional shortening occurs based on their observations. Specifically, the authors propose that myosin-dependent shape fluctuations at the apical surface of the cells surrounding the vertical interface cause one of the flanking tricellular junctions (vertices) to slide within the plasma membrane to shorten that interface. They further observe that E-cadherin and myosin undergo cycles of accumulation and enrichment at the vertices that correlate with the fluctuations in cell area and propose that high E-cad levels (stimulated by myosin accumulation) stabilize the vertex after a period of sliding.

The reviewers are impressed with the quantitative analyses performed in this study and find many of the key observations supporting the authors' model to be clear and compelling. The reviewers further feel that the newly proposed mechanism may have broad implications for our understanding of epithelial morphogenesis in a variety of systems. However, there are aspects of the data/model that need to be clarified, and some of the authors' conclusions are insufficiently supported by the data shown. We have the following suggestions to improve the manuscript.

Essential revisions:

In the list of essential revisions below, only points 11 and 12 require new experiments. Points 4, 7, 8, and 9 require further analyses of existing movies. The remaining points are all simply revisions to the text.

1) *eLife* requires that the title and/or abstract provide a clear indication of the biological system under investigation. Please revise your Title and/or Abstract to mention both *Drosophila* and germ band extension.

2) Subsection “Movement of cell vertices is physically coupled in the radial direction” – The following statement is problematic: "the inward movement of vertices connected by a contracting interface should show evidence of mechanical coupling". The movement of a vertex is defined by the force balance between the 3 junctions connected to the vertex. Even if the vertices connected by contracting interfaces are pulled inward by the same contractility, if each vertex is pulled outward by different forces, the dynamics of 2 connected vertices could be different. Indeed, two recent studies showed that the medial myosin at the vicinity of vertex contributes to the horizontal junction elongation during GBE, suggesting the pulling by the neighboring cells contribute to the movement of vertex (Collinet, 2015; Yu, 2016). Thus, 2 connected vertices don't need to synchronize their movement, and this observation cannot be used as evidence to dismiss the line tension-driven model.

3) Figure 2B, B' – The authors should clearly state what the criteria are for imposing a grey or blue vertical bar over a region of the graph. In particular, it is confusing why the second blue bar in panel B' seems to sit over a local peak instead of a consistent downward slope.

4) Figure 2D-F – The cellular dynamics shown in Video 2 make a compelling case that vertex sliding can occur during GBE. However, the reviewers raised three concerns about these data:

- The authors should clarify how often the vertex sliding mechanism contributes to the shortening of vertical interfaces. The authors state that Figure 2F shows that "analysis of the lengths of all contracting and adjacent interfaces throughout GBE demonstrated that this behavior is a systematic component of T1-associated vertex movements" (subsection “Vertices slide independently one another”). However, the methods state that the analysis was limited to interfaces that took 5 minutes to contract (subsection “Vertex sliding shown through sum of contracting and transverse interfaces”). What percentage of total contracting interfaces do the data represent? If you looked at interfaces that took 3 minutes to contract, or 7 minutes, would the result be the same?

- The authors should show an example of vertex sliding that includes a full T1 transition. Video 2 only shows the partial shorting of an AP boundary. Moreover, the contraction slowed down over time (Figure 2E). Thus, this example may show junctional remodeling, but may not be a T1 transition.

- The authors should explicitly test whether there are differences between the contraction of junctions oriented from dorsal to ventral and the expansion of junctions oriented along the length of the embryo. More specifically, it is important to test the extent of junctional sliding (and ratchet) in DV and AP directions, since under their model such a difference is likely to be the major driver of morphogenesis. In addition, both expansion and contraction cycles should be studied in the context of junctional contractions (e.g. Figure 2). The model depends on Myosin II and E-cadherin fixing the junction in place at the end of a contraction cycle. However, the data are shown in Figure 5 and Figure 6, as if contraction and expansion events are symmetrical. Additionally, the text states "E-cadherin intensity peaks just before cell area is in its most contracted state". This is confusing.

5) Figure 3 – The observation that the flanking transverse interfaces grow as a vertical interface shrinks is interesting, and the data seem convincing. However, this observation appears to be at odds with their recent paper (Jewett et al., 2017). In this paper, the authors propose that Rab35 acts as part of the ratcheting mechanism that shrinks vertical interfaces by promoting the endocytic removal of plasma membrane from this location. Please discuss and try to reconcile these two models.

6) Figure 4 – The authors observe that E-cad intensities peak with vertex stabilization. They then use chlorpromazine to artificially stabilize E-cad levels and see that this disrupts vertex sliding and conclude: "Thus, stabilizing E-cadherin prevents vertex sliding and uncouples the ratcheted motion of individual vertices from the motive oscillations in cell area" (subsection “E-cadherin intensities are in phase with area oscillations and peak with vertex stabilization”). This conclusion is too strong based on the data shown. Injecting a general inhibitor of Clathrin-mediated endocytosis like chlorpromazine has far more effects on the cell than just stabilizing E-cad. For example, the Jewett et al. paper shows that the Rab35 compartments along vertical interfaces fail to terminate under these conditions. The authors should point out that a defect in Rab35 dynamics could provide an alternate explanation for their chlorpromazine results.

7) Figure 4B and Figure 6B – To help the reader, the two cross correlation analyses should be expanded to include additional data. First, please show the still images and associated graphs for still images and associated graphs for Figure 4B and 6B. Second, please include the cross-correlation between apical area and cadherin/myosin intensity, as well as the rates of change (note that the main text fails to state what is being correlated). This would help to show what happens at the end of each contraction cycle. How long do E-cadherin and Myosin II remain at the vertex to bias the expansion movement? It would also make it clear whether or not myosin drives the accumulation of E-cadherin at vertices as suggested. Finally, the authors should include data showing how junctional length variations like those shown in Figure 2B correlate with fluctuations in apical area.

8) Figure 5 – It is important to test whether the apical constriction events are isotropic along the two axes of the embryo. If they are not, then some of the measurements presented are misleading, since apical area changes represent changes of different magnitude in AP and DV axes.

9) Figure 6 – The authors show that myosin accumulates at the junctions just before E-cad. The reviewers listed the following concerns about these data:

- The dynamics of MyoII that are shown are very different from those previously presented (Rauzi, 2010), where MyoII moves toward the cell-cell interfaces and accumulates in a polarized manner. Moreover, MyoII distribution shown in Figure 6A (white dotted rectangle), with E-cad accumulation at the vertex, does not show a polarized MyoII accumulation. In the same data, the AP boundaries with high junctional MyoII aren't associated with high accumulation of E-cad at vertices (see the AP boundary close to the bottom left corner of the white dotted rectangle in Figure 6A). This indicates that there could be two types of AP boundary. It would be very important and interesting to report what fraction of AP boundaries shows (1) high junctional MyoII and low (?) E-cad intensity at the vertex, and (2) low junctional MyoII (high vertex MyoII) and high E-cad intensity at the vertex.

- The section of the results that discusses this figure is titled "Myosin II directs phasic E-cadherin enrichment at cell vertices". This statement is insufficiently supported by the data shown. The correlation between myosin and E-cad accumulating at the junctions is clear, but the experiment involving the rok inhibitor are too pleiotropic to draw such a specific conclusion, both because multiple myosin-dependent processes are being affected in these cells, and because this drug also affects more proteins that just myosin.

- Based on the model, one would predict (as has been seen in the notum) that a loss of Myosin II, increasing junctional length fluctuations. Is this the case following Y27632 treatment?

10) Discussion – the authors need to discuss more fully how their data fit with the wealth of information already known about GBE. How might the AP patterning system feed into the proposed mechanism? In lines 318-319, the authors state: "While higher line tensions at AP interfaces clearly exist and direct distinct aspects of intercalary cell behaviors […]". What behaviors do they direct? Can vertex sliding lead to the formation of rosettes? How does this model fit with the role of basal protrusions in cell intercalation? With the role of Rab35? Is vertex sliding the only mechanism driving cell intercalation during GBE, or is it one of many? Please help the non-expert reader connect the dots.

11) To directly test their assertion that increased E-cad levels at the vertices inhibit their sliding, the authors should vary E-cad dosage (e.g. haploid to triploid).

12) The analyses carried out with myosin-mCherry constructs should be repeated with myosin-GFP constructs, as the mCherry constructs often give misleading results.

[Editors' note: further revisions were requested prior to acceptance, as described below.]

Thank you for resubmitting your work entitled "Vertex sliding drives intercalation by radial coupling of adhesion and actomyosin networks during *Drosophila* germband extension" for further consideration at *eLife*. Your revised article has been favorably evaluated by a Guest Editor and Didier Stainier as the Senior Editor.

The manuscript has been improved but there are some remaining issues that need to be addressed before acceptance, as outlined below:

1) The new experiment performed in *sqh* mutant embryos (Figure 7—figure supplement 1) is not mentioned in the Results section of the manuscript. Please add this information.

2) There are at least two references that are cited in the text that do not appear in the reference list. I noticed Takeda, 2018 and Jewett, 2017, but there may be others. Please do a careful accounting to be sure that the reference list is correct.

3) There is a typo in the Discussion section "primary" should be "primarily".

4) There is some red text in the legends for Figure 5—figure supplement 1 and Figure 6—figure supplement 1. Please convert this text to black.

---

## [Author Response]

Essential revisions:In the list of essential revisions below, only points 11 and 12 require new experiments. Points 4, 7, 8, and 9 require further analyses of existing movies. The remaining points are all simply revisions to the text.1) eLife requires that the title and/or abstract provide a clear indication of the biological system under investigation. Please revise your Title and/or Abstract to mention both Drosophila and germ band extension.

We have edited the title to match *eLife*’s requirements – “Vertex sliding drives intercalation by radial coupling of adhesion and actomyosin networks during *Drosophila* germ band extension.”

2) Subsection “Movement of cell vertices is physically coupled in the radial direction” – The following statement is problematic: "the inward movement of vertices connected by a contracting interface should show evidence of mechanical coupling". The movement of a vertex is defined by the force balance between the 3 junctions connected to the vertex. Even if the vertices connected by contracting interfaces are pulled inward by the same contractility, if each vertex is pulled outward by different forces, the dynamics of 2 connected vertices could be different. Indeed, two recent studies showed that the medial myosin at the vicinity of vertex contributes to the horizontal junction elongation during GBE, suggesting the pulling by the neighboring cells contribute to the movement of vertex (Collinet, 2015; Yu, 2016). Thus, 2 connected vertices don't need to synchronize their movement, and this observation cannot be used as evidence to dismiss the line tension-driven model.

We agree — this is a deeply interesting point to consider, and we hope our manuscript will have the opportunity to contribute to the debate. We have added to our Discussion section along some of these lines, but there are a number of interesting points in this comment. To our knowledge, all truly physics-based models currently proposed rely on interface-spanning line tensions. We also agree that in these physical models the movement of a vertex will depend on the forces exerted by the three (or more) connecting interfaces as well as radially-directed forces. Although this means there are multiple influences, if there are interface-spanning line tensions at contracting interfaces it is an absolute physical principle that there must be coupling between the two adjacent vertices (i.e., there is an equal and opposite force component on the two vertices toward the middle of the interface). It is therefore quite compelling that the only observable coupling is radial coupling. We examined this data with a variety of time windows and other criteria to eliminate potential hysteresis. However, we also agree that there are molecular models that do not rely on line tensions and include a new Discussion section on this. We believe that since line tension-based understandings of cell intercalation have been so prevalent that our findings are critical data for the field to grapple with. The cited work from the Lecuit and Fernandez-Gonzalez labs is highly interesting and important studies, but concerns the growth of horizontal, T3 interfaces and not the contracting T1 interfaces that involve proposed contractile line tension and which most GBE studies have been concerned with. We think these T3 behaviors are very intriguing but are currently beyond the scope of this work.

3) Figure 2 B, B' – The authors should clearly state what the criteria are for imposing a grey or blue vertical bar over a region of the graph. In particular, it is confusing why the second blue bar in panel B' seems to sit over a local peak instead of a consistent downward slope.

We agree – the shaded regions were manually defined based on highly active periods of inward or outward motion and these criteria are now stated in the figure legend. We also went through another 10-15 examples and selected a better vertex pair example for the figure. Two additional examples are shown in Author Response Image 1 to illustrate that the chosen example is representative and depict the strength of radial coupling.

**Author Response Image 1 respfig1:** 

4) Figure 2D-F – The cellular dynamics shown in Video 2 make a compelling case that vertex sliding can occur during GBE. However, the reviewers raised three concerns about these data:- The authors should clarify how often the vertex sliding mechanism contributes to the shortening of vertical interfaces. The authors state that Figure 2F shows that "analysis of the lengths of all contracting and adjacent interfaces throughout GBE demonstrated that this behavior is a systematic component of T1-associated vertex movements" (subsection “Vertices slide independently one another”). However, the methods state that the analysis was limited to interfaces that took 5 minutes to contract (subsection “Vertex sliding shown through sum of contracting and transverse interfaces”). What percentage of total contracting interfaces do the data represent? If you looked at interfaces that took 3 minutes to contract, or 7 minutes, would the result be the same?

Thank you for catching this – this was phrased badly in the original text as the analysis was actually performed on all contracting interfaces with lifetimes of at least 5 minutes. This represents ~92% of interfaces, and we have edited the methods to reflect this as, “This sum was performed on the last 5 minutes of all contracting interfaces that had a lifetime of at least 5 minutes.

- The authors should show an example of vertex sliding that includes a full T1 transition. Video 2 only shows the partial shorting of an AP boundary. Moreover, the contraction slowed down over time (Figure 2E). Thus, this example may show junctional remodeling, but may not be a T1 transition.

Yes, we concentrated the video on a single sliding event to emphasize this behavior, as vertices will go through multiple sliding events/steps to complete a full T1 contraction. However, we have also added Figure 2G (image panels) and Figure 2H (interface length plots) that show vertex sliding for a full T1 transition.

- The authors should explicitly test whether there are differences between the contraction of junctions oriented from dorsal to ventral and the expansion of junctions oriented along the length of the embryo. More specifically, it is important to test the extent of junctional sliding (and ratchet) in DV and AP directions, since under their model such a difference is likely to be the major driver of morphogenesis. In addition, both expansion and contraction cycles should be studied in the context of junctional contractions (e.g. Figure 2). The model depends on Myosin II and E-cadherin fixing the junction in place at the end of a contraction cycle. However, the data are shown in Figure 5 and Figure 6, as if contraction and expansion events are symmetrical. Additionally, the text states "E-cadherin intensity peaks just before cell area is in its most contracted state". This is confusing.

We are a little unclear on what is meant by this point. Figure 5 shows that vertex displacements are correlated with area oscillations. However, while area oscillations and cyclic changes in vertex E-cadherin and Myosin II are indeed symmetrical (Figure 3E, Figure 4B, Figure 5H and Figure 6C’), vertex displacements are not (Figure 3C and Figure 5C,E). What these data demonstrate is that cycles of area expansion/contraction are used as periods when cell vertices can displace in processive, ratcheted events. These vertex displacements are asymmetrical and produce interface shortening specifically in an AP shortening direction (the tangential motion that is plotted in Figure 5). The expansion and contraction cycles are represented in the -180 to +180 degree measurements Figure 5, but we welcome further clarification of this comment. On the other hand, if the reviewer is referring to comparing the role of sliding in T1 contraction (in DV direction) and T3 elongation (in AP direction), we agree that these T3 behaviors are very intriguing, but the manuscript became too complicated to also include T3 behaviors. We are currently planning on reporting these results in a separate work.

5) Figure 3 – The observation that the flanking transverse interfaces grow as a vertical interface shrinks is interesting, and the data seem convincing. However, this observation appears to be at odds with their recent paper (Jewett et al., 2017). In this paper, the authors propose that Rab35 acts as part of the ratcheting mechanism that shrinks vertical interfaces by promoting the endocytic removal of plasma membrane from this location. Please discuss and try to reconcile these two models.

Yes, this is a nice insight – we have expanded on this point in the new Discussion section. There are several potential explanations, but we believe the most likely one is that the data in this study indicates that the endocytic pathways reported on in Levayer et al., 2011 and Jewett et al., 2017 act to alter the balance of adhesion on either side of a vertex through the uptake of E-cadherin adhesion molecules. When E-cadherin becomes cyclically depleted at cell vertices, this would permit the sliding of the vertex in productive directions in response. However, our data shows that the combined length of T1 and transverse interfaces remains largely unchanged, and this suggests that the overall uptake of the plasma membrane and lipids does not limit total cell perimeters and interface lengths, but rather these endocytic pathways primarily affect the uptake of E-cadherin and the balance of adhesion complex function. This was one of the possibilities we discussed in the Rab35 manuscript, but we were not able to follow E-cadherin endocytosis well enough to discriminate between these possibilities. We have added these ideas into the new Discussion section, and this will be a very interesting question to delve into further in future studies.

6) Figure 4 – The authors observe that E-cad intensities peak with vertex stabilization. They then use chlorpromazine to artificially stabilize E-cad levels and see that this disrupts vertex sliding and conclude: "Thus, stabilizing E-cadherin prevents vertex sliding and uncouples the ratcheted motion of individual vertices from the motive oscillations in cell area" (subsection “E-cadherin intensities are in phase with area oscillations and peak with vertex stabilization”). This conclusion is too strong based on the data shown. Injecting a general inhibitor of Clathrin-mediated endocytosis like chlorpromazine has far more effects on the cell than just stabilizing E-cad. For example, the Jewett et al. paper shows that the Rab35 compartments along vertical interfaces fail to terminate under these conditions. The authors should point out that a defect in Rab35 dynamics could provide an alternate explanation for their chlorpromazine results.

Yes, that is a good point – to address this we have examined E-cadherin overexpressing embryos. By increasing the overall genetic dosage of E-cadherin, we tested whether vertex behaviors would be altered. Indeed, in Ubiquitin (promoter)-E-cadherin:GFP; Gap43:mCh embryos, vertex displacements and ratcheting are reduced and trend in the direction of chlorpromazine-injected embryos, although, as expected, certainly not to the same degree (Figure 4—figure supplement 2C-C’’). This supports a likely requirement for periods of decreased E-cadherin during vertex movements.

7) Figure 4B and Figure 6B – To help the reader, the two cross correlation analyses should be expanded to include additional data. First, please show the still images and associated graphs for still images and associated graphs for Figure 4B and 6B. Second, please include the cross-correlation between apical area and E-cadherin/Myosin intensity, as well as the rates of change (note that the main text fails to state what is being correlated). This would help to show what happens at the end of each contraction cycle. How long do E-cadherin and Myosin II remain at the vertex to bias the expansion movement? It would also make it clear whether or not Myosin drives the accumulation of E-cadherin at vertices as suggested. Finally, the authors should include data showing how junctional length variations like those shown in Figure 2B correlate with fluctuations in apical area.

We have added this data. Figure 6—figure supplement 1C-C’’’ shows image panels in both E-cadherin and Myosin channels of a vertex (C), the intensity ratios of the vertex in both Myosin and E-cadherin (C’), the rate of change of the intensity ratios (C’’) and the cross-correlation of the rates (C’’’). To Figure 3—figure supplement 1B we have added image panels and a plot showing cell area and interface length as an example of length ratcheting. To Figure 4—figure supplement 1A-A’’’ we have added plots of E-cadherin vertex intensity and apical cell area (A’) of the vertex shown in (A), their rates of change (A’’), and the cross-correlation of the rates of change (A’’’).

8) Figure 5 – It is important to test whether the apical constriction events are isotropic along the two axes of the embryo. If they are not, then some of the measurements presented are misleading, since apical area changes represent changes of different magnitude in AP and DV axes.

Our data shows that the oscillation magnitudes in the AP and DV directions are very similar in our data sets (see Author response image 2), and, importantly, we have also not found any evidence for a significant phase shift of oscillations between AP and DV directions. If we understand them correctly, the reviewer raises a valid point in that it would be theoretically possible to create T1 interface contraction mechanism through an anisotropic radial constriction system alone, where all forces are directed only into the radial direction, but with different magnitudes in the AP vs DV directions; we agree, and this is why we are explicitly measuring the tangential vertex displacements (Figure 5C,D) to differentiate this motion from radial displacements that could arise from constriction anisotropy.

**Author response image 2. respfig2:** 

9) Figure 6 – The authors show that myosin accumulates at the junctions just before E-cad. The reviewers listed the following concerns about these data:- The dynamics of MyoII that are shown are very different from those previously presented (Rauzi 2010 Nature), where MyoII moves toward the cell-cell interfaces and accumulates in a polarized manner. Moreover, MyoII distribution shown in Figure 6A (white dotted rectangle), with E-cad accumulation at the vertex, does not show a polarized MyoII accumulation. In the same data, the AP boundaries with high junctional MyoII aren't associated with high accumulation of E-cad at vertices (see the AP boundary close to the bottom left corner of the white dotted rectangle in Figure 6A). This indicates that there could be two types of AP boundary. It would be very important and interesting to report what fraction of AP boundaries shows (1) high junctional MyoII and low (?) E-cad intensity at the vertex, and (2) low junctional MyoII (high vertex MyoII) and high E-cad intensity at the vertex.

We believe the medial Myosin II dynamics observed are fairly similar to those shown by Rauzi et al., and did not mean to indicate the Myosin II flows solely to cell vertices. Similar to Rauzi et al., we observe medial Myosin moving towards the cell periphery, and have added new quantitation (Figure 6E) showing that this Myosin has an essentially equal probability to merge with a vertex or interface. This data supports the reported relationship between medial Myosin and vertex enrichments but does not indicate that Myosin II is only also associated with cell vertices. We also mention aspects of this in the Discussion section, as it is possible that interface-associated Myosin, in additional to medial and vertex Myosin, plays a role in local force generation that may bias vertex displacements.

We also completed analysis to assess if there are two distinct Myo/E-cad populations (Author response image 3). E-cad vertex enrichment in E:cad:GFP; Myosin:mCh movies is grouped consistently around vertex enrichments of ~ 3.0, while junctional Myosin II distributions possess a broader range. These data do not appear to provide evidence of distinct high E-cad vertex:low junctional Myo (and vice versa) populations.

**Author response image 3. respfig3:** 

- The section of the results that discusses this figure is titled "Myosin II directs phasic E-cadherin enrichment at cell vertices". This statement is insufficiently supported by the data shown. The correlation between Myosin and E-cad accumulating at the junctions is clear, but the experiment involving the rok inhibitor are too pleiotropic to draw such a specific conclusion, both because multiple myosin-dependent processes are being affected in these cells, and because this drug also affects more proteins that just Myosin.

This is a valid point and we have performed additional experiments to address this (Figure 7—figure supplement1A-B) – a similar loss of vertex-associated E-cadherin occurs in *sqh*^AX^ mutant embryos (*sqh* is the gene encoding the RLC of Myosin II in *Drosophila*), supporting the functional conclusions of the effect of Y-27632 on Myosin II and E-cadherin.

- Based on the model, one would predict (as has been seen in the notum) that a loss of Myosin II, increasing junctional length fluctuations. Is this the case following Y27632 treatment?

This is an interesting comment – we do not believe that our model predicts increasing junctional length fluctuations in Y-27632 treated embryos. We agree that although decreasing Myosin II reduces vertex E-cad, which could in principle promote mobility, our results also suggest that the main driver of mobility are oscillations in area driven by medial Myosin II populations (we make mention of this in subsection “Myosin II directs phasic E-cadherin enrichment at cell vertices”). As the function of these populations is also reduced, we do not believe that increased vertex mobility would be observed. To look at this more closely, we have used junction length step detection and show that frequency (B) and magnitude (A) of junctional length steps is reduced in the Y27632 injected embryos. Furthermore, the heatmaps in Figures 7H and 7I demonstrate that both forward and backward stepping is reduced in magnitude in the Y-27632 treated embryos.

**Author response image 4. respfig4:** 

10) Discussion – the authors need to discuss more fully how their data fit with the wealth of information already known about GBE. How might the AP patterning system feed into the proposed mechanism? In lines 318-319, the authors state: "While higher line tensions at AP interfaces clearly exist and direct distinct aspects of intercalary cell behaviors […] ". What behaviors do they direct? Can vertex sliding lead to the formation of rosettes? How does this model fit with the role of basal protrusions in cell intercalation? With the role of Rab35? Is vertex sliding the only mechanism driving cell intercalation during GBE, or is it one of many? Please help the non-expert reader connect the dots.

Yes, we agree with this comment – We have extensively re-written the Discussion to fit our work into the larger context of GBE-related work. There are a few points raised by the reviewer that we do not address, for example, since our work is generally concerned with apical events and does not address events that are basal to the adherens junction, although this is an interesting topic for a review/future studies.

11) To directly test their assertion that increased E-cad levels at the vertices inhibit their sliding, the authors should vary E-cad dosage (e.g. haploid to triploid).

This was a nice idea (see comment #6 as well) – to address this we have examined Ubi-E-cadherin overexpressing embryos. In Ubiquitin (promoter)-E-cadherin:GFP; Gap43:mCh embryos, vertex displacements and ratcheting are reduced and trend in the direction of chlorpromazine-injected embryos, although certainly not to the same high degree (see Figure 4—figure supplement 2C-C’’). This supports a likely requirement for periods of decreased E-cadherin during vertex movements.

*12) The analyses carried out with myosin-mCherry constructs should be repeated with myosin-GFP constructs, as the mCherry constructs often give misleading results*.

We repeated the analysis with a Zipper-GFP protein trap line which has GFP inserted at the endogenous locus. Zipper-GFP; Gap43:mCherry embryos, shown in Figure 6—figure supplement1B-B’’, E, possess similar enrichment of Myosin II at vertices (B-B’’) and similar recruitment of Myosin to the vertices of AP junctions (E).

[Editors' note: further revisions were requested prior to acceptance, as described below.]

The manuscript has been improved but there are some remaining issues that need to be addressed before acceptance, as outlined below:1) The new experiment performed in sqh^AX^ mutant embryos (Figure 7—figure supplement 1) is not mentioned in the Results section of the manuscript. Please add this information.

A description of these results is now included.

2) There are at least two references that are cited in the text that do not appear in the reference list. I noticed Takeda, 2018 and Jewett, 2017, but there may be others. Please do a careful accounting to be sure that the reference list is correct.

We have carefully gone through the reference list and ensured that all references are listed (sorry, we had done this previously but must have exchanged a wrong version).

3) There is a typo in the Discussion section. "primary" should be "primarily".

Thank you for catching this (some very careful reading!).

4) There is some red text in the legends for Figure 5—figure supplement 1 and Figure 6—figure supplement 1. Please convert this text to black.

We have converted the text to black. (this had also been corrected previously but was not saved in the submitted version – thank you for pointing it out).